# Characterizing steroid hormone receptor chromatin binding landscapes in male and female breast cancer

Tesa M. Severson[1], Yongsoo Kim[1,2], Stacey E.P. Joosten[2], Karianne Schuurman[2], Petra van der Groep[3], Cathy B. Moelans[3], Natalie D. ter Hoeve[3], Quirine F. Manson[3], John W. Martens [4,5,6], Carolien H.M. van Deurzen[4,5], Ellis Barbe[4,7], Ingrid Hedenfalk[8], Peter Bult[9], Vincent T.H.B.M. Smit[4,10], Sabine C. Linn[3,11,12], Paul J. van Diest[3], Lodewyk Wessels[1,6,13] & Wilbert Zwart[2]

Male breast cancer (MBC) is rare and poorly characterized. Like the female counterpart, most MBCs are hormonally driven, but relapse after hormonal treatment is also noted. The pan-hormonal action of steroid hormonal receptors, including estrogen receptor alpha (ERα), androgen receptor (AR), progesterone receptor (PR), and glucocorticoid receptor (GR) in this understudied tumor type remains wholly unexamined. This study reveals genomic cross-talk of steroid hormone receptor action and interplay in human tumors, here in the context of MBC, in relation to the female disease and patient outcome. Here we report the characterization of human breast tumors of both genders for cistromic make-up of hormonal regulation in human tumors, revealing genome-wide chromatin binding landscapes of ERα, AR, PR, GR, FOXA1, and GATA3 and enhancer-enriched histone mark H3K4me1. We integrate these data with transcriptomics to reveal gender-selective and genomic location-specific hormone receptor actions, which associate with survival in MBC patients.

[1] Division of Molecular Carcinogenesis, Oncode Institute, Netherlands Cancer Institute, 1066 CX Amsterdam, The Netherlands. [2] Division of Oncogenomics, Oncode Institute, Netherlands Cancer Institute, 1066 CX Amsterdam, The Netherlands. [3] Department of Pathology, University Medical Center, 3584 CX Utrecht, The Netherlands. [4] Dutch Breast Cancer Research Group, (BOOG Study Center), 1076 CV Amsterdam, The Netherlands. [5] Department of Medical Oncology, Erasmus Medical Center, 3015 CE Rotterdam, The Netherlands. [6] Cancer Genomics Netherlands, Universiteitsweg 100, 3584 CG Utrecht, The Netherlands. [7] Department of Pathology, VU Medical Center (VUMC), 1081 HV Amsterdam, The Netherlands. [8] Division of Oncology and Pathology, Lund University, 221 00 Lund, Sweden. [9] Department of Pathology, Radboud University Medical Center, 6525 GA Nijmegen, The Netherlands. [10] Department of Pathology, Leiden University Medical Centre, 2333 ZA Leiden, The Netherlands. [11] Division of Medical Oncology, Netherlands Cancer Institute, Amsterdam, 1066 CX, The Netherlands. [12] Division of Molecular Pathology, Netherlands Cancer Institute, Amsterdam, 1066 CX, The Netherlands. [13] Faculty of EEMCS, Delft University of Technology, 2628 CD Delft, The Netherlands. Tesa M. Severson, Yongsoo Kim and Stacey E.P. Joosten contributed equally to this work. Correspondence and requests for materials should be addressed to W.Z. (email: w.zwart@nki.nl)

Breast cancer in men is rare and largely understudied. Male breast cancer (MBC) accounts for around 1% of all 1.67 million breast cancers diagnosed each year, worldwide[1]. As compared to the female counterpart, MBC is a distinct disease regarding clinicopathological features (e.g., age of onset and frequency of hormone receptor positivity[2]) and molecular features (e.g., frequency of *BRCA2* mutation[3] and gene expression subtypes[4]).

The majority of male (and female) breast cancers are hormonally driven[5], where ERα genomic action dictates transcriptional programs that drive tumor cell proliferation[6]. Analogous to the female counterpart, male breast cancers are treated with endocrine therapies (such as tamoxifen) to block ERα transcriptional activity, yet relapse after hormonal treatment has also been noted[2,7]. Even though the genomic action of ERα in MBC remains completely elusive, multiple reports have studied ERα genomics in the female disease. ERα-DNA binding profiles in tumors are dynamically affected by endocrine therapeutics[8] and can differentiate female patients on outcome[9,10]. Cell line studies revealed ERα-DNA binding and ERα-driven transcriptional activation and cell proliferation to depend on its pioneer factors, including FOXA1[11] and GATA3[12].

Apart from ERα, other steroid hormone receptors are expressed in breast cancer as well, including androgen receptor (AR)[13], progesterone receptor (PR)[14], and glucocorticoid receptor (GR)[14]. AR expression is frequently observed in most male (and female) breast cancers[13–16], although its role in breast cancer is poorly understood[17]. AR activation in breast cancer cells facilitates downstream gene expression that drives tumorigenesis in a similar manner to ERα[16]. This tumorigenic action of AR is most extensively studied in prostate cancer[18,19], where differential AR-DNA binding profiles can classify prostate cancer patients on outcome[20–22].

AR and PR are favorable prognostic markers in female breast cancer (FBC)[23,24]. In addition, PR has recently been shown to modulate ERα-DNA binding, directly reprogramming ERα-driven transcriptional programs[25]. GR expression has been associated with FOXA1 and GATA3 expression in ERα-positive FBC, and is associated with a favorable outcome in this patient population[26]. Its functional role in breast cancer in relation to other steroid hormone receptors is poorly characterized. Cumulatively, these data illuminate the likely interplay between different steroid hormone receptors in breast cancer. Although ERα cistromics has previously been studied in female breast tumors[9,10], and its interplay with transcription factors has been reported in cell lines[10,11,15,27–31], all these transcription factors have never been profiled together in a single study in human breast tumors.

We have characterized DNA binding of six different hormone-related transcription factors in an understudied field of human pathophysiology: male breast cancer. Through multidimensional genomic data integration on the level of transcription factor binding, copy number cistrome profiling (using off-target sequencing reads from ChIP-seq data)[32], transcriptomics and the enhancer enriched histone mark H3K4me1, we present a first comprehensive overview of male breast cancer, which we compared with the female counterpart. This comprehensive overview reveals gender-selective and genomic location-specific hormone receptor action, which associate with survival in MBC.

## Results

**Steroid hormone receptor profiling in male breast cancer.** We aimed to generate a compendium of (epi)genomic, transcriptomic and clinical data for 49 ERα–positive MBC samples to better characterize the molecular makeup of this disease. To determine the chromatin binding landscape of ERα in relation to steroid hormone receptors AR, PR, and GR and its pioneer factors FOXA1 and GATA3, we performed ChIP-seq analyses in clinical specimens from patients who did not receive any therapy prior to surgery. These results were integrated with gene expression data and compared with female breast cancer and cell line ChIP-seq data (Fig. 1a, Supplementary Fig. 1). Samples were selected randomly for ChIP-seq of different factors. In this pioneering work, each transcription factor was profiled in MBC for the first time (30 ERα ChIP-seq datastreams and ≥7 samples/factor for other factors with the exception of GATA3 and PR, with 3 and 4 samples, respectively). To position these results into epigenetic context, H3K4me1 was included as active enhancer marker[33]. We generated RNA-seq data for the series, used to classify samples on subtypes related to outcome; M1 (poor) and M2 (good)[4] (Supplementary Fig. 1). Finally, we used copy number data (detected using off-target sequencing reads)[32] and RNA-seq data to perform integrative clustering (IntClust) classification, which was previously associated with FBC prognostication (Supplementary Fig. 1)[34]. As expected, IntClust classifications and intrinsic subtypes (based on immunohistochemistry) were enriched for ERα-positive classifications (29/30 and 28/28, respectively). Clinical data, such as number of positive lymph nodes at diagnosis and survival status were included for male (Supplementary Fig. 1 and Supplementary Table 1) and female patients (Supplementary Table 2). Missing clinical data are indicated in Supplementary Table 1. We identified bound regions (peaks) for each factor: ERα (biological replicate in Supplementary Fig. 2), FOXA1, AR, GR, PR, and GATA3 (ChIP-seq validated by ChIP-QPCR in Supplementary Fig. 3) using validated antibodies (Supplementary Fig. 4A, B), shown at two well-known ERα bound regions in FBC: loci *RARA* and *GREB1* (Fig. 1b). In this principal examination of SHRs, FOXA1 and GATA3 in a tumor series, we observed these two ERα-regions were bound by all other factors. This is reminiscent of FBC transcription factors, such as GATA3, FOXA1, RARα, which have been found to be bound in the same regions[35]. These findings were confirmed on a genome-wide scale, where all ERα sites were considered bound in ≥50% of tumors in which we find co-binding by all other factors tested (Fig. 1c). In accordance with reports in cell lines[11,25,29,36], all factors studied mainly bind intronic and distal intergenic regions (Fig. 1d). DNA motif analysis revealed self-preference for all factors (Fig. 1e, Supplementary Fig. 5) except for GATA3 as was reported previously[12].

**Interplay of ERα with other SHRs and pioneer factors.** We have shown ERα binding sites have considerable overlap with other factors studied. Next, overlap of ERα with the other steroid hormone receptors (Fig. 2a, left) or pioneer factors FOXA1 and GATA3 (Fig. 2a, right) was studied. All sites shared for individual factors: ERα (15 out of 30 samples), AR (5 out of 10 samples), FOXA1 (3 out of 7 samples), PR (2 out of 4 samples), GR (3 out of 7 samples) and GATA3 (1 out of 3 samples). AR and GR have virtually no unique binding regions, while selective sites for ERα and PR are identified. When examining ERα, FOXA1 and GATA3, we identified selective sites in each. Strikingly, 71% of FOXA1 sites were FOXA1-selective while ER and GATA3-selective sites were 42% and 33%, respectively. Next, we counted the reads in the union of sites for all factors (Fig. 2a) and computed the correlation score for each sample (Pearson correlation coefficient), which was represented in a correlation matrix (Fig. 2b). Contrary to what is described for ERα/FOXA1 behavior in FBC cell lines[11], we observed FOXA1 profiles do not correlate with other profiles, while all SHRs and GATA3 cluster together. Most notably, a strong similarity in genomic profiles of ERα with GR and AR was found. Practically all AR sites were co-occupied

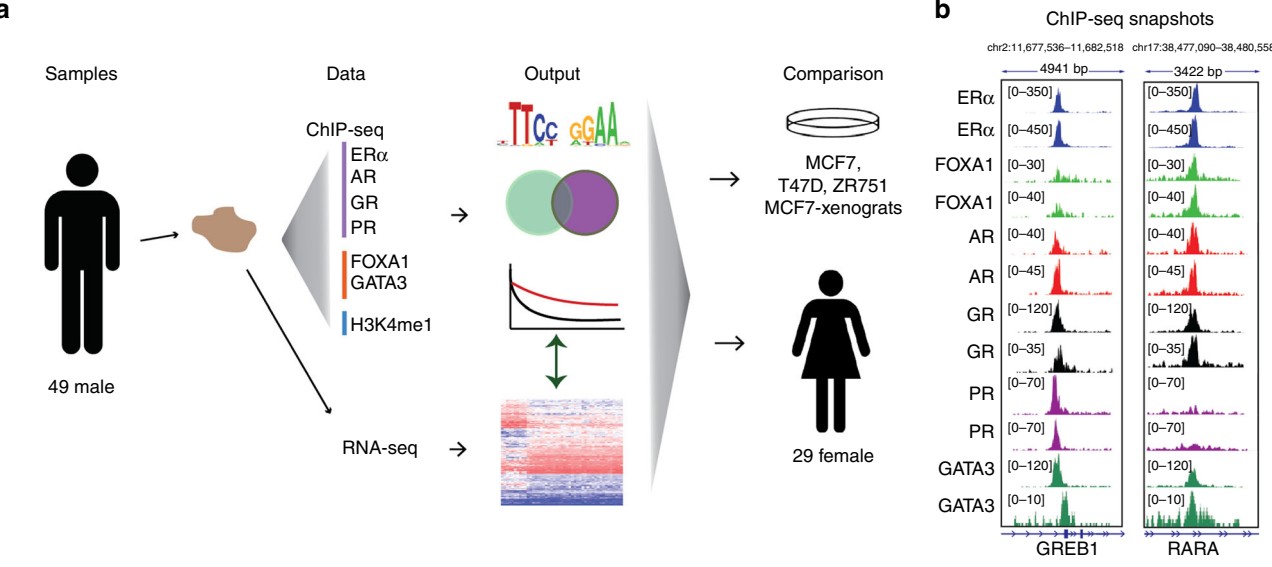

**a** Samples, Data, Output, Comparison diagram

**b** ChIP-seq snapshots

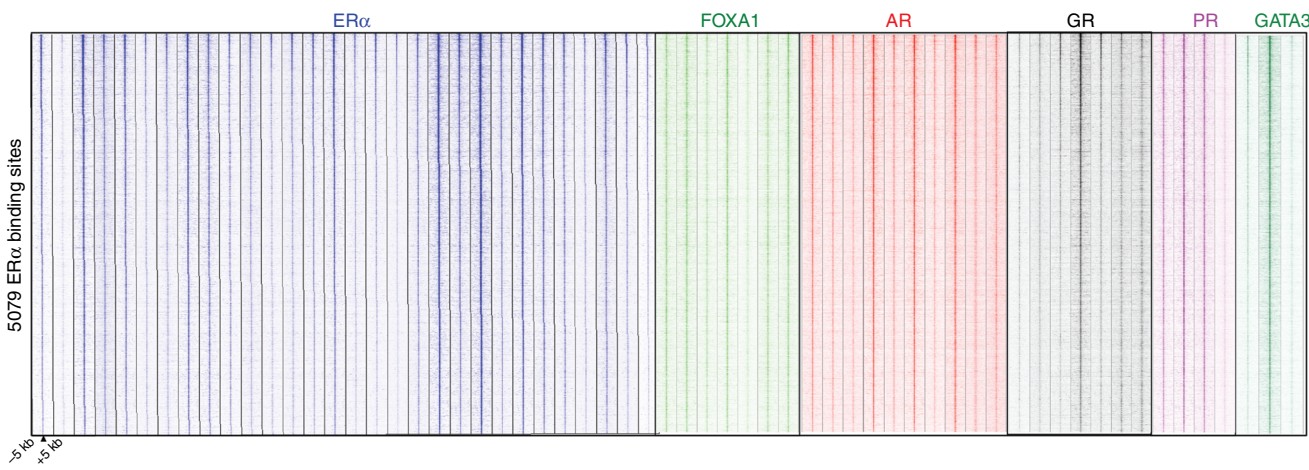

**c** ChIP-seq of different factors at ERα binding sites

5079 ERα binding sites

ERα   FOXA1   AR   GR   PR   GATA3

−5 kb  +5 kb

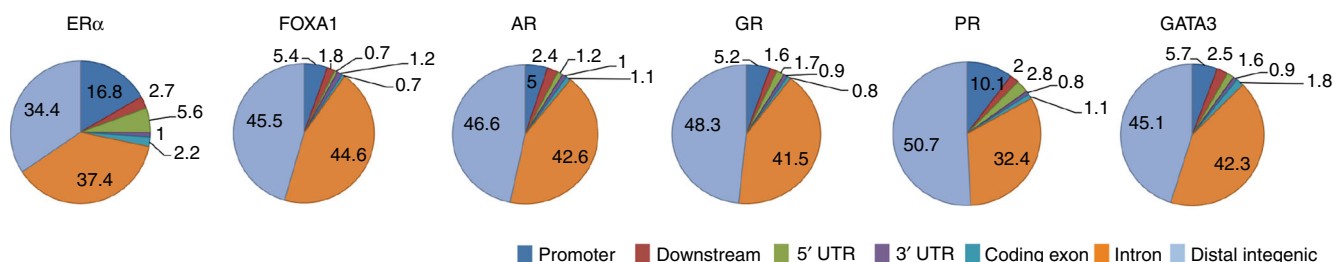

**d** Genomic distribution of binding regions for different factors

ERα | FOXA1 | AR | GR | PR | GATA3

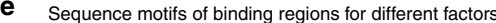

Promoter   Downstream   5′ UTR   3′ UTR   Coding exon   Intron   Distal integenic

**e** Sequence motifs of binding regions for different factors

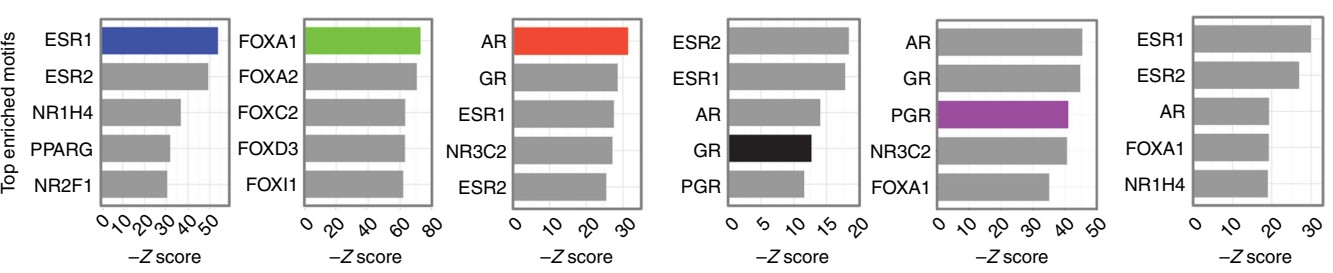

Top enriched motifs

−Z score

with ERα (Fig. 2c), which was also seen in female breast tumor and MCF7 cells (Supplementary Fig. 6A, B).

MBC is ERα-driven[37,38]. Therefore, a one-to-one comparison of binding sites of ERα with each factor was performed (Fig. 2c, d and Supplementary Fig. 7), where only samples were analyzed in which both ERα and the other factor were profiled (peak selection as stated above and in figure legends). A significant proportion of ERα binding sites overlap with AR and GR sites as seen in cell line data[39,40] which we confirmed in MBC (Fig. 2c, d). Interestingly, although PR modulates ERα binding in FBC[25], many PR sites in MBC were devoid of ERα (46%). These findings were confirmed in DNA binding correlation matrices (Fig. 2e). ERα and AR show clustering within patient rather than between factors (Fig. 2e), indicating a stronger correlation between factors within the same tumor as compared to the same factor between tumors.

A dominant dogma of ERα biology purports ERα binding is dependent on its pioneering factor FOXA1, with 95% of binding found at enhancer regions[11,29,30,41,42]. In line with literature[11], we find ~50% overlap between ERα and FOXA1 in cell lines, where sites enriched for FOXA1 or ERα (Fig. 3a, b and Supplementary Fig. 8) were largely found in intronic and distal intergenic regions (Fig. 3c). These findings are in stark contrast to observations in both male and female breast cancers, in which ERα sites devoid of FOXA1 were strongly promoter-enriched (Fig. 3a–c), suggesting the model systems currently used do not adequately capture the genomic distributions of ERα found in clinical samples. In contrast to FOXA1-enriched sites, sites selectively occupied by ERα were weaker for active enhancer mark H3K4me1[33] (Supplementary Fig. 9A, B). Interestingly, GATA3 was found at both the FOXA1-enriched and ERα-enriched sites (Supplementary Fig. 9A). Motifs at ERα selective sites are related to ESR1 and devoid of forkhead motifs (Supplementary Fig. 10), which is in contrast to the total of ERα sites (Supplementary Fig. 5).

**MBC subtypes differ in hormone receptor action.** Having characterized MBC with respect to SHRs, GATA3 and FOXA1 DNA binding, we next performed gene expression analyses in these samples ($n = 46$). In order to assess underlying ERα binding patterns between published MBC intrinsic subtypes M1 and M2[4], we first confirmed M1/M2 subtype clustering using our RNA-seq data set using only subtype genes (Fig. 4a). Supporting our hypothesis that ERα function may deviate between M1 and M2 subtypes, we found 1395 differential ERα DNA binding sites (Fig. 4b). Analogous analyses were not performed for other factors than ERα, since ChIP-seq datastreams were not sufficiently powered to represent both the M1 and M2 subtypes (Supplementary Fig. 1). With available datastreams, we confirmed the occupancy of FOXA1, AR, GR, PR, and GATA3 at these differential ER α DNA binding sites (Fig. 4c). M1- and M2-specific

sites were comparable in genomic location and motif usage (Fig. 4d, e and Supplementary Fig. 11). Genes with proximal binding sites (<20 kb or within the gene body) were subsequently examined for molecular and biological associations using pathway analysis (IPA, Qiagen). Both ERα ChIP-seq associated M1/M2 genes and gene expression-based M1/M2 genes strongly associated with ERα pathway indicators as expected, though some additional regulators are specifically found in the previously reported expression-based classification, such as ERBB2 and KRAS (Fig. 4f). Interestingly, among canonical pathways, AR signaling was the only hormonal signaling pathway more associated with the ERα binding based genes compared to the gene expression-based genes (Fig. 4g), in line with the strong overlap of AR/ERα binding in these tumors (Figs. 2, 4c).

**Comparing genomics of ERα and FOXA1 between genders.** As ERα is the key driver and therapeutic target in both genders, we compared ERα chromatin binding in female (17 from Ross-Innes et al.[10], 9 from Jansen et al.[9] and 10 generated in-house) and male (30 generated in-house) breast tumors (Fig. 5a), along with its pioneer factor FOXA1 ($n = 7$ for both genders) (Fig. 5b). Interestingly, no clear differences in ERα and FOXA1 binding was found between genders, on the level of peak overlap ratio (Fig. 5a–d) or relative read counts in peaks (Fig. 5e, f). For ERα and FOXA1 sites found in ≥50% of male tumors, signal was observed in female samples at comparable intensity (Fig. 5e, f, Supplementary Fig. 12A, B, Supplementary Fig. 13), and vice versa. Furthermore, motif enrichment at ERα and FOXA1 sites was highly comparable between genders (Fig. 5g, h). While clear clustering was observed for ERα between (male and female) tumors and cell lines (Fig. 5a), no separation on gender was observed for any of the factors studied in an integrative analysis (Supplementary Fig. 14), as well as separately for all factors studied (Fig. 5a, b, Supplementary Fig. 15). ERα sites that classify FBC on outcome[10] were used to predict male outcome ($k$-nearest neighbor classifier; Methods section), and a weak but similar trend of ERα signal strength was observed in these sites (Fig. 5i). However, overall survival (OS) was not significantly different between the two groups of male patients (Supplementary Fig. 16). These results suggest that although the vast majority of ERα and FOXA1 sites are conserved between breast cancers from both genders, ERα sites indicative for outcome in FBC may not be applicable in the male disease.

**Genomic profiles of ERα and FOXA1 stratify MBC patients on outcome.** Since prognostic ERα sites in female tumors do not seem to be indicative for male patient outcome in our MBC series, we analyzed binding sites of ERα and pioneer factor FOXA1 in MBC for outcome prediction. Analogous analyses for ERα/GATA3 were not performed due to insufficient power due to

---

**Fig. 1** Study schematic and steroid hormone receptor binding in male breast cancer. **a** A graphic visualization of the study design. The male silhouette was from Wikipedia (https://en.wikipedia.org/wiki/File:Male_Bathroom_Symbol.png). The female silhouette was from Wikipedia (https://commons.wikimedia.org/wiki/File:Toilet_women.svg), from a collection commissioned by the United States Department of Transportation, designed by AIGA http://www.aiga.org/content.cfm/symbolsigns, and converted into SVG by Wikipedia-user Lateiner. The image is used under a CC-BY 2.5 license. **b** Genome browser snapshot, depicting two known ERα bound regions with read counts for 2 random samples chosen for each factor. Genomic coordinates and read counts are indicated above. **c** Heatmaps depicting peak intensity in primary tumors for 30 ERα (blue), 7 FOXA1 (light green), 10 AR (red), 7 GR (black), 4 PR (purple), and 3 GATA3 (dark green) binding events (±5 kb from the peak center (triangle)). 5079 ERα sites were determined using the consensus ERα binding sites identified in at least 50% of patients (15 out of 30 samples). **d** Pie charts depicting genomic distributions for the consensus binding sites of each of the factors: AR (shared in 5 out of 10 samples), FOXA1 (3 out of 7 samples), PR (2 out of 4 samples), GR (3 out of 7 samples), and GATA3 (1 out of 3 samples). **e** Negative Z-score of the top 5 sequence motifs for binding sites of each factor depicted as a barplot. Colored (non-gray) bar represents the target factor's sequence motif

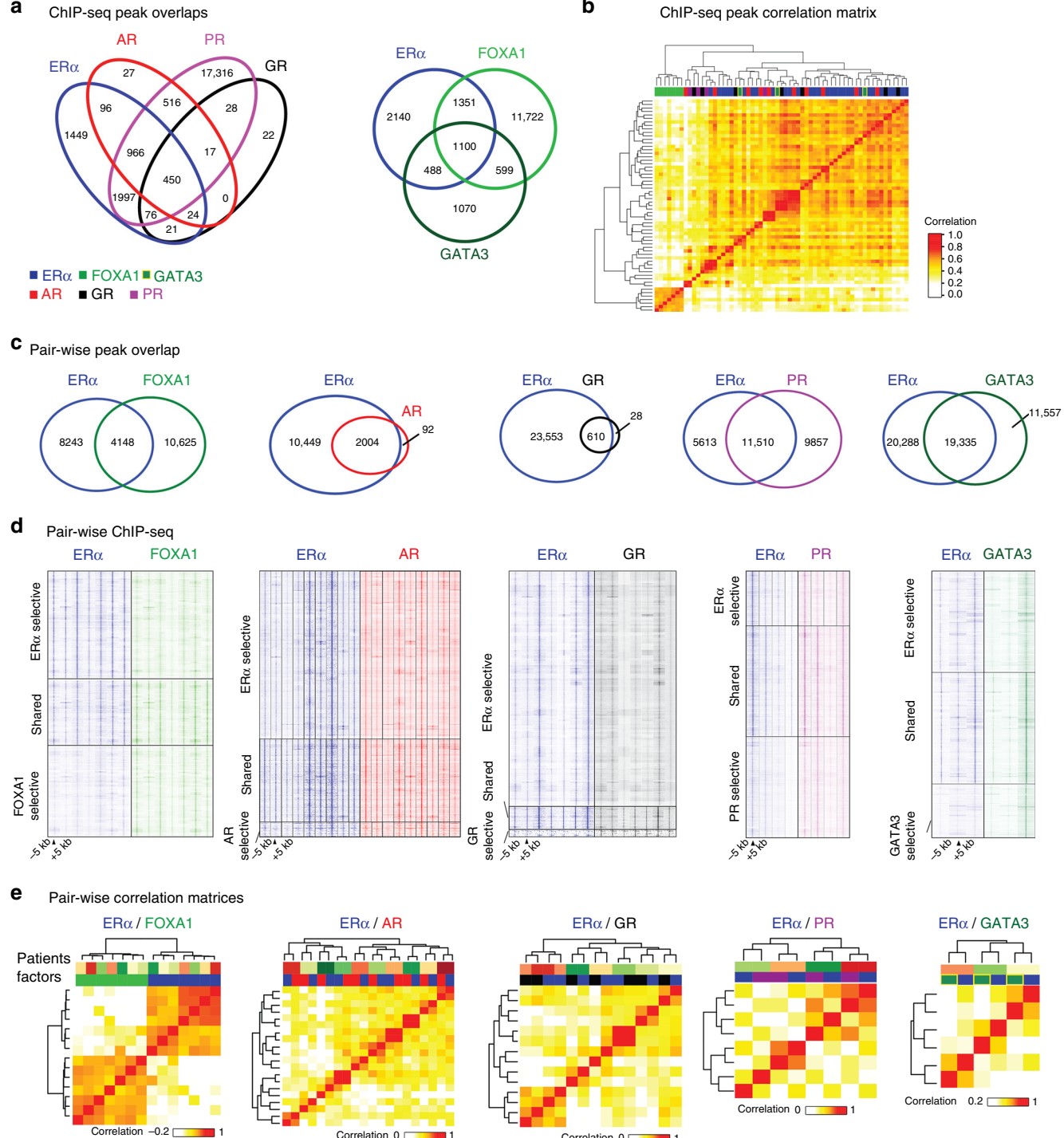

**Fig. 2** Multifactorial ChIP-seq data integration. **a** Venn diagram depicting the overlap of ERα with steroid hormone receptors (left) and overlap of ERα with FOXA1 and GATA3 (right). **b** Unsupervised clustering correlation matrix of bound regions among all factors. Top bar indicates the factors: ERα (blue), AR (red), PR (purple), GR (black), FOXA1 (light green), and GATA3 (dark green/orange border). Pearson correlation is depicted from 0 (white) to 1 (dark green). Sites used to construct the matrix are the union of all consensus binding sites of the factors. **c** Individual Venn diagrams depicting the pairwise overlap of ERα with each factor. Consensus binding sites for each factor were used (Fig. 1d), and overlap with ERα sites from the same tumors was assessed, taking the same threshold. **d** Heatmaps indicating the binding peak intensity for each combination in (**c**). In each panel, ERα (left) and factor (e.g., FOXA1, right) ChIP-seq data are depicted. Shown are peaks selectively observed for ERα (top), selectively observed for other factors (bottom), or shared between them (middle). **e** Correlation heatmap for ChIP-seq data sets between ERα and the other factors. Colors in top bar indicate patient (top) and factor (bottom)

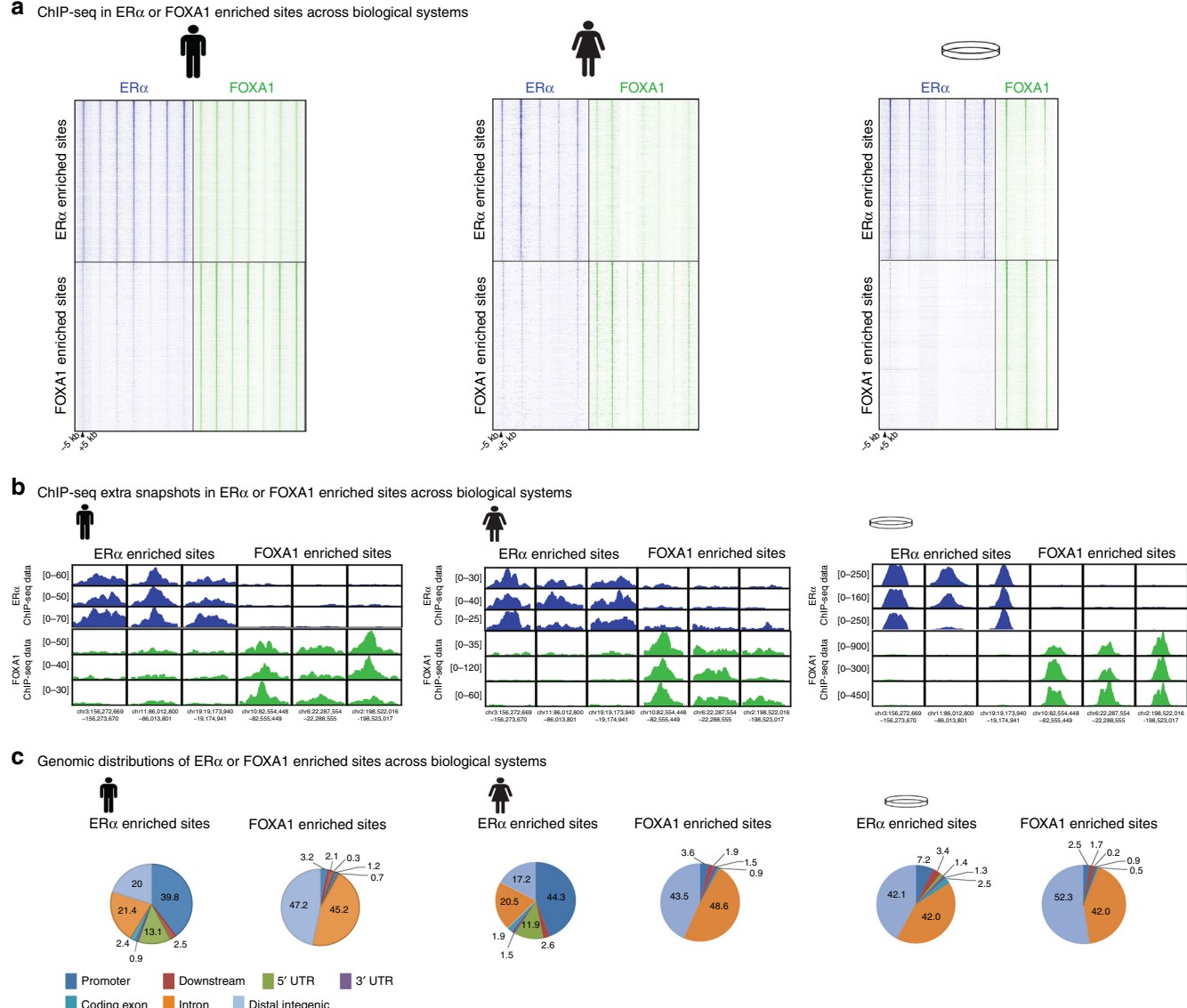

**Fig. 3** ERα- and FOXA1-enriched binding sites across biological systems. **a** Heatmaps indicating the binding peak intensity in sites, differentially enriched between ERα (blue) and FOXA1 (green). Left, middle and right columns indicate male tumor, female tumor and female cell line data, respectively. The male silhouette was from Wikipedia (https://en.wikipedia.org/wiki/File:Male_Bathroom_Symbol.png). The female silhouette was from Wikipedia (https://commons.wikimedia.org/wiki/File:Toilet_women.svg), from a collection commissioned by the United States Department of Transportation, designed by AIGA http://www.aiga.org/content.cfm/symbolsigns, and converted into SVG by Wikipedia-user Lateiner. The image is used under a CC-BY 2.5 license. **b** Genome browser snapshot depicting sites differentially bound by ERα or FOXA1 across biological systems. Genomic coordinates and read count are indicated. **c** Pie charts indicating genomic distribution of ERα- (left) and FOXA1-enriched (right) sites across biological systems

small ($n = 3$) sample size. Based on lymph node status, indicative for overall survival (Supplementary Fig. 17), 365 ERα and 470 FOXA1 sites differed (Fig. 6a). Differential ERα and FOXA1 sites between patient subgroups were coupled to proximal genes (<20 kb or within the gene body). Unsupervised hierarchical clustering revealed clusters dominated by either of M1 or M2 subtypes, both in our cohort (Fig. 6b) and a validation set[4] (Fig. 6c; $n = 66$ patients). We performed logistic regression with elastic net regularization[43] to construct a supervised binary classification model by which predictive gene signatures could be identified, which was trained in our cohort and tested in the validation cohort. Both ERα- and FOXA1-based classifiers captured predictive features, which were outperformed by the union of both classifier gene lists (Fig. 6d, e). A bootstrapping analysis confirmed that comparable performance is rarely achieved with random gene sets ($p = 0.013$, one-tailed test with bootstrapped

performance distribution; Methods section; Supplementary Fig. 18). Dividing patients into two groups of equal size based on the signature (high-risk and row-risk group of LN-status) significantly classified patients on distant metastasis free survival (DMFS; $p = 0.048$, log-rank test; Fig. 6f; Methods section), which was marginally significant in a multivariate Cox analysis including LN-status ($p = 0.066$, Cox proportional hazards test). The gene expression signature is a linear combination of gene expression levels, in which 14 contributing genes classify MBC on outcome (Fig. 6g). Cumulatively, we show that global ERα and FOXA1 chromatin binding selectivity reveals gender-specific prognostic features that successfully classify MBC patients on survival.

## Discussion

This work has characterized the DNA binding landscape of ERα in male breast cancer, along with its pioneer factors FOXA1,

GATA3, and enhancer-enriched histone modification H3K4me1. In addition, we present the first set of DNA binding data in breast tumor specimens for other members of the steroid hormone receptor family: AR, GR, and PR. Our findings have indicated that the majority of ERα binding sites in both male and female breast tumors are FOXA1-independent and are found at active promoter regions, indicating a novel and unexpected mode of ERα function. These results are in stark contrast to cell line-based studies that illustrated the majority of ERα sites shared with FOXA1, mainly found at enhancers[11]. Our findings highlight the

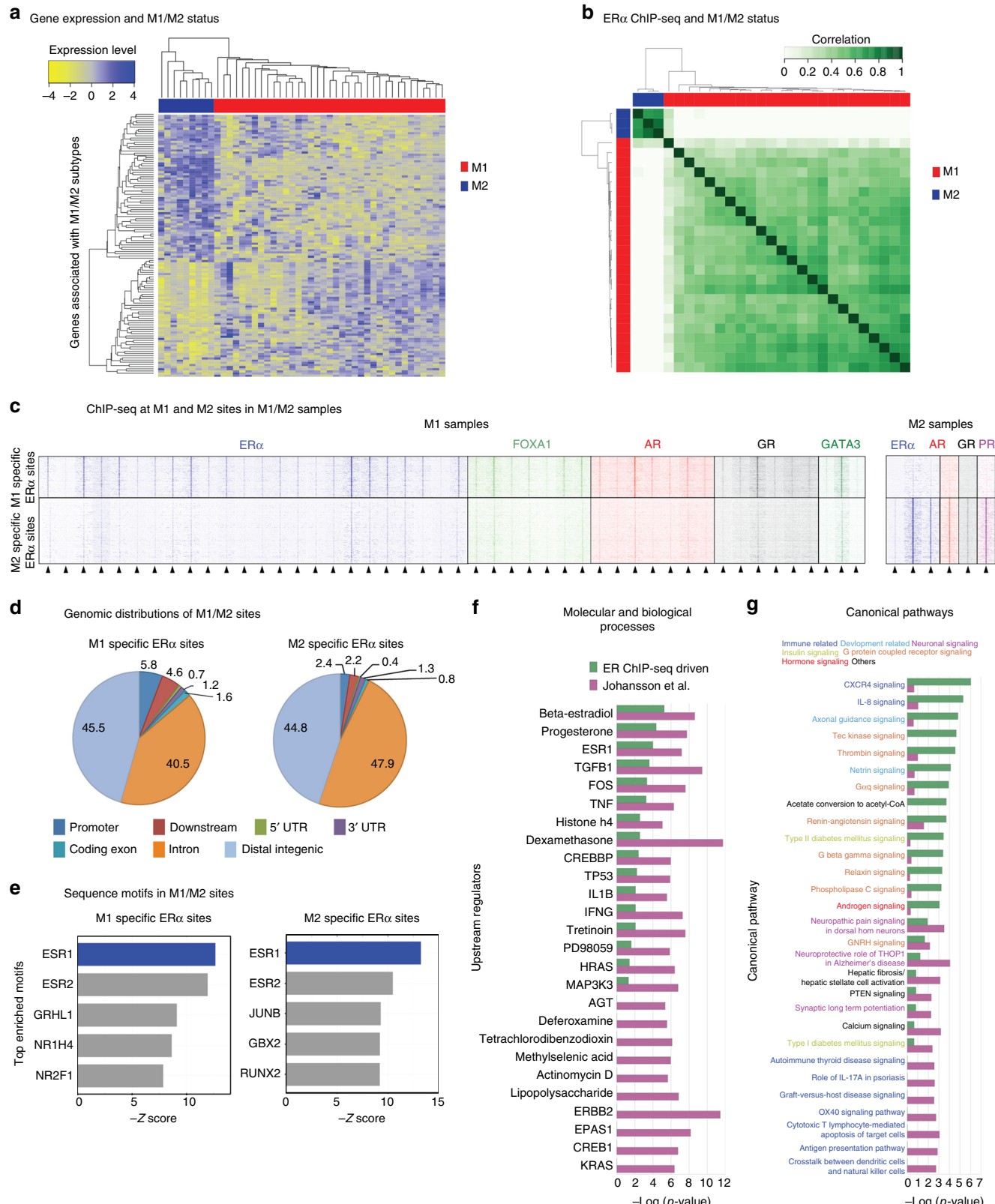

necessity to address transcription factor functioning in the physiological context of human tissue, rather than limiting analyses to cell line models.

Our data reveal that genomic functions of ERα and AR in male breast tumors are largely overlapping, which strongly co-localized with GR and PR at the same regions. Even though many sites for GATA3, FOXA1, and PR were not shared with ERα, both AR and GR show virtually no unique binding sites with respect to ERα binding. AR has been shown to compensate for ERα in ERα-/AR+ female breast cancers[40,44], however the biological interaction between ERα and AR is relatively unknown in both FBC and MBC[45]. The observed genomic overlap of steroid hormone receptor binding profiles is likely due to the close sequence homology of DNA binding domains between all steroid hormone receptors, which warrants potential competition between them in DNA binding. Alternatively, genomic overlap of SHR binding profiles may be the consequence of 'tethering', in which factors associate to the DNA indirectly through complex formation with DNA-bound factors[35], as was recently described for ERα and PR[25]. Such genomic convergence of steroid hormone receptor action in tumors may provide a novel starting point for pharmaceutical intervention strategies, yielding direct biological rationale for the use of small molecule therapeutics to target AR (e.g., clinicaltrials.gov NCT01990209), GR or PR in hormone receptor-positive breast cancer. As MBC is a rare cancer with limited numbers of available tumors for genomic studies, ChIP-seq analyses for some factors including PR and GATA3 were performed on relatively low number of tumors. To focus our analyses on the most-robust peaks and thus minimizing potential impact of patient heterogeneity, for all SHRs we only considered peaks that were found in around 50% of patient samples. This could be considered a rather conservative approach compared to other tissue-derived ChIP-seq papers, where the union of all peaks[46] or peaks identified in at least 2 out of 21 tumors[10] were used as consensus for analyses. Nonetheless, results for PR and GATA3 still warrant validation in larger cohorts.

Although, we found the vast majority of ERα sites to be shared between male and female breast cancers, ERα sites that are associated with patient outcome appeared gender-selective. In line with these results, genomic selectivity of combinatorial steroid hormone receptor action is associated with the gender-specific intrinsic MBC subtypes M1 and M2. While these data suggest ERα function may be driving these subtypes, causality can only be illustrated when cell line models, organoids or patient-derived xenografts are available for mechanistic studies. As the most clinically relevant observation, we have identified distinct genomic signatures of ERα action, which selectively and exclusively classifies MBC patients on outcome. With differential binding of ERα and FOXA1 as a guide, we developed a gene expression signature that is significantly associated with DMFS in MBC patients. The union of genes under differential control of ERα and FOXA1 jointly classify patients on outcome, and it remains to be determined which transcription factor is facilitated by FOXA1 at these sites. The MLL3 histone methyltransferase

may represent one candidate to be tested in future studies based on the published FOXA1 and MLL3 interaction in FBC cells[47]. The 14 genes classifier we identified may be of added value as male breast cancer-specific prognostic classifier, but further validation of these results would be needed. Furthermore, small molecule inhibitors are available for a number of the 14 genes represented in the classifier, such as CAMKK2 (STO-609)[48], CAPN9[49], BACE2[50], and TNFSF11 (aka RANKL)[51], and future studies could further elucidate whether these inhibitors may be applicable in the treatment of male breast cancer.

With this, we present the first comprehensive genomic overview of shared and unique features of four steroid hormone receptors in human cancer, with outcome prediction. By studying MBC, gender-selective features of ERα action were identified with potentially direct clinical implications, revealing the first biology-driven biomarker for outcome prediction in this highly understudied cancer-type.

## Methods

**Tumor specimens**. In this study, primary male and female breast tumors were used, none of whom received neoadjuvant endocrine therapy. Male breast cancer patients received surgery at the Netherlands Cancer Institute-Antoni van Leeuwenhoek (NKI-AVL; Amsterdam, the Netherlands), University Medical Center Utrecht (UMCU; Utrecht, the Netherlands), Vrije Universiteit Medical Center (VUMC; Amsterdam, the Netherlands), Radboud University Medical Center (RadboudUMC; Nijmegen, the Netherlands), University Medical Center Groningen (UMCG; Groningen, The Netherlands), Leiden University Medical Center (LUMC; Leiden, the Netherlands), and Erasmus Medical Center (ErasmusMC; Rotterdam, the Netherlands).

Female breast cancer patients received surgery at the Netherlands Cancer Institute-Antoni van Leeuwenhoek (NKI-AVL; Amsterdam, the Netherlands). Tumor content and immunohistochemical analyses were assessed by pathological examination. For clinicopathological parameters, see Supplementary Tables 1 (male tumors) and 2 (female tumors). Local medical ethical authorities at abovementioned centers approved of the collection protocols. All samples were from anonymous left-over material, which would be discarded otherwise. Anonymized, coded leftover material which is not traced back to the patient and therefore does not interfere with care and/or prognosis, under strict requirements can be used without written informed consent according to Dutch legislation on Secondary Use[52].

**ChIP-seq and antibody validations**. Tissue was processed as described previously[53] with a few adaptations. In short, tissue was defrosted and crosslinked in solution A (50 mM Hepes, 100 mM NaCl, 1 mM EDTA, 0.5 mM EGTA, pH = 7.4) containing 2 mM DSG, incubated for 25 min at room temperature while rotating. After 25 min formaldehyde was added to 1% final and incubated another 20 min at room temperature with rotation. Samples were quenched by adding a surplus of 0.2 M glycine, pelleted by centrifugation (5'@4000 r.c.f. at 4 °C), washed with cold PBS and mechanically disrupted in cold PBS using a pellet pestle (Sigma). The PicoBioruptor (Diagenode) was used for sonication. For ChIP, antibodies were used to detect ERα (sc-543, Santa Cruz), AR (sc-816, Santa Cruz), FOXA1 (sc-6554, Santa Cruz), PR (sc-7208, Santa Cruz), GR (12041 S lot 3, Cell Signaling Technology), GATA3 (sc-268, Santa Cruz), and H3K4me1 (ab8895, AbCam). Immunoprecipitated DNA was prepared for Illumina multiplex-sequencing with 10 samples per lane at 65 bp single end. For steroid hormone receptor ChIPs except GR, 5 μg of antibody and 50 μl dynabeads (Invitrogen) were used, for GR 7.5 μg of antibody and 75 μl dynabeads were used, and for FOXA1 and H3K4me1, 4 μg of antibody and 40 μl magnetic beads were used. ChIP-QPCR was performed to validate ChIP-seq data for ERα, GR, AR, and FOXA1. For QPCR, relative enrichment of the RARA enhancer (chr:1738478661-38478809) (primers: 5′-GCTGGGTCCTCTGGCTGTTC-3′ (FWD) and 5′-

**Fig. 4** Steroid hormone receptor binding differs between male breast cancer subtypes. **a** Unsupervised hierarchical clustering of RNA-seq of 46 male breast tumors, using the M1/M2 classifier genes defined by Johansson et al.[4] Standardized gene expression value is denoted as the row Z-score and plotted in high expression (blue) and low expression (yellow) scale. Top row indicates M1 (red) or M2 (blue) classification. **b** Correlation plots of significantly differentially bound regions for ERα between 24 M1 (red) and 3 M2 (blue) tumors. Pearson correlation is depicted from 0 (white) to 1 (dark green) for all panels. **c** Heatmaps for all factors with known M1/M2 subtype classification, depicting raw signal intensities at differential bound ERα sites between M1 and M2. **d** Genomic distribution for differentially ERα-bound sites by M1 and M2 subtypes. **e** Bar plots indicating top 5 sequence motifs enriched in M1 and M2 differentially ERα-bound sites. **f** Ingenuity Pathway Analysis of Upstream Regulators and their p-values identified in both ERα binding site associated and Johannson et al.[4] gene expression based driven M1/M2 genes. **g** Ingenuity Pathway Analysis of Canonical Pathways and their p-values identified in both ERα binding site associated and Johannson et al.[4] gene expression based driven M1/M2 genes

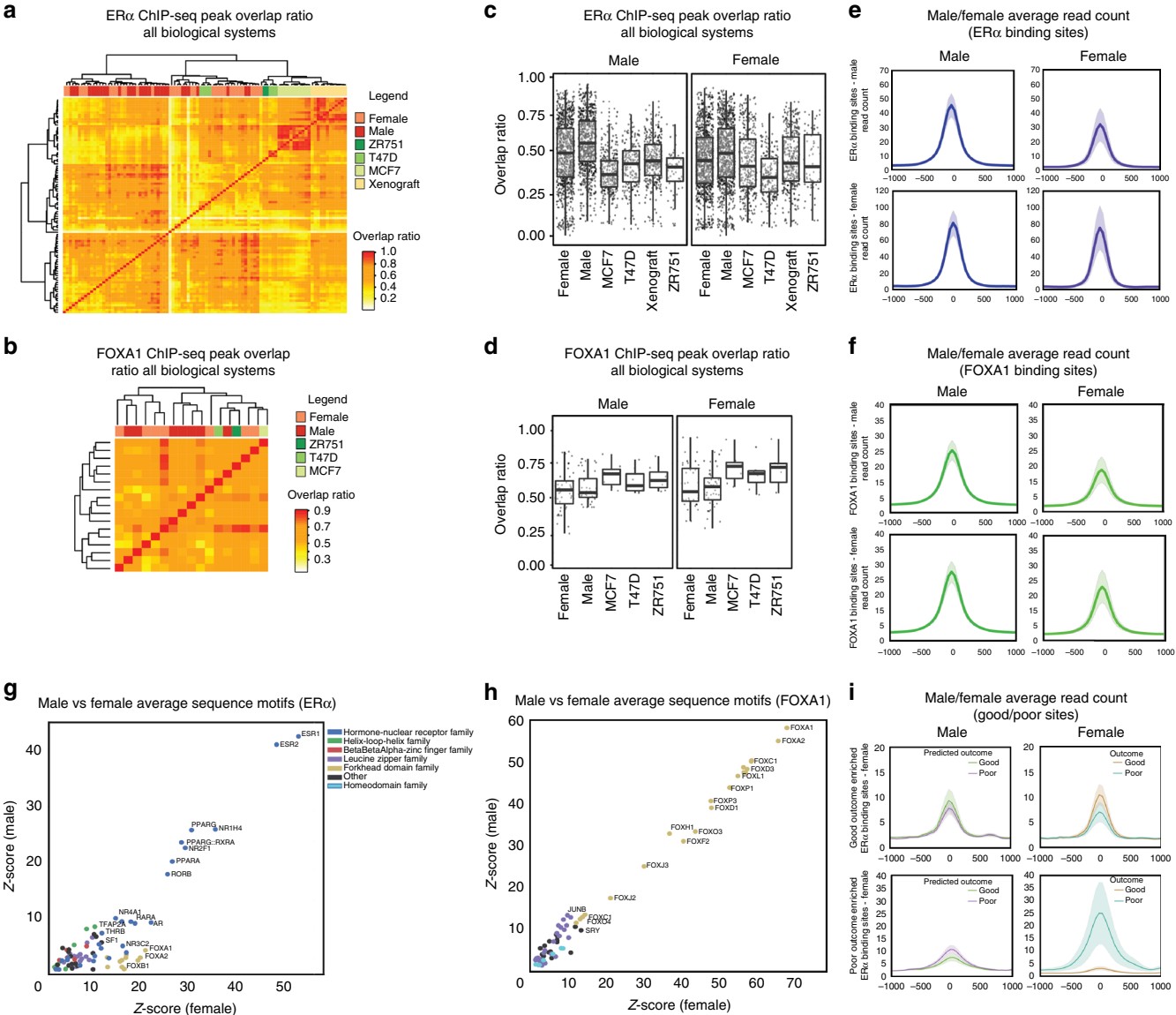

**Fig. 5** Strong conservation of ERα and FOXA1 binding between genders. **a** Correlation plot of ERα bound regions across all biological systems. Top row indicates female tumors (salmon), male tumors (red), and female cell lines ZR751 (dark green), T47D (green), MCF7 (light green) and MCF7 xenograft (yellow) classification. Overlap ratio is depicted from 0 (white) to 1 (red). **b** Boxplots depicting all overlap values for ERα bound regions in male (left) and female (right) samples. **c** Correlation plot of FOXA1 bound regions across all biological systems. Top row indicates female tumors (salmon), male tumors (red), and cell lines ZR751 (dark green), T47D (green) and MCF7 (light green) classification. Overlap ratio is depicted from 0.3 (white) to 1 (red). **d** Boxplots depicting all overlap values for FOXA1 bound regions in male (left) and female (right) samples. **e** Average ERα read count profiles for male ERα binding sites (50% consensus, top panel) and female ERα binding sites (50% consensus, bottom panel) in male (left) and female (right) datasets. 75% confidence interval of read count profiles are indicated with shading. **f** Average FOXA1 read count profiles for male FOXA1 binding sites (50% consensus, top panel) and female FOXA1 binding sites (50% consensus, bottom panel) in male (left) and female (right) data sets. 75% confidence interval of average profiles are indicated with shading. **g** Scatter plot depicting Z-scores of significantly enriched motifs at ERα binding sites in male (y-axis) and female (x-axis) tumors. **h** Scatter plot depicting Z-scores of significantly enriched motifs at FOXA1 binding sites in male (y-axis) and female (x-axis) tumors. **i** Average ERα read count profiles of male (left) and female (right) tumors at the differential ERα binding sites (±5 kb from the peak center) that can discriminate female outcome (top—good outcome enriched; bottom—poor outcome enriched). Patients are grouped based on outcome where indicated color is used for each group. 75% confidence interval of average profiles are indicated with shading

CCGGGATAAAGCCACTCCAA-3′ (REV)) over a negative control region (primers: 5′-TGCCACACACCAGTGACTTT-3′ (FWD) and 5′-ACAGCCA-GAAGCTCCAAAAA-3′ (REV) was normalized over input, and plotted against the log(count/million) measurement from the same region from the MQ20 filtered aligned ChIP-seq file. SHR antibody specificity was validated by immunohistochemistry on U2OS cells transiently transfected with any of the SHRs of interest (Supplementary Fig. 3A). Antibody used for FOXA1 ChIP-seq also detects FOXA2[54] but only FOXA1 is expressed in these tumors (Supplementary Fig. 3B). Specificity for the antibody used for H3K4me1 ChIP-seq (ab8895)[55,56] and GATA3

(sc-268) has been validated by others[57,58]. Publically available ChIP-seq data sets used are listed in Supplementary Table 3.

**Immunohistochemistry**. Immunohistochemistry staining for ER, PR, and AR was performed as previously described[8]. Immunohistochemistry of other factors was performed on a BenchMark Ultra autostainer (GATA3 and FOXA1) or Discovery Ultra autostainer (GR). Briefly, paraffin sections were cut at 3 μm, heated and deparaffinized in the instrument with EZ prep solution (Ventana Medical Systems). FOXA1 was detected using clone 2F83 (1/100,000 dilution, 16 min at RT, Seven

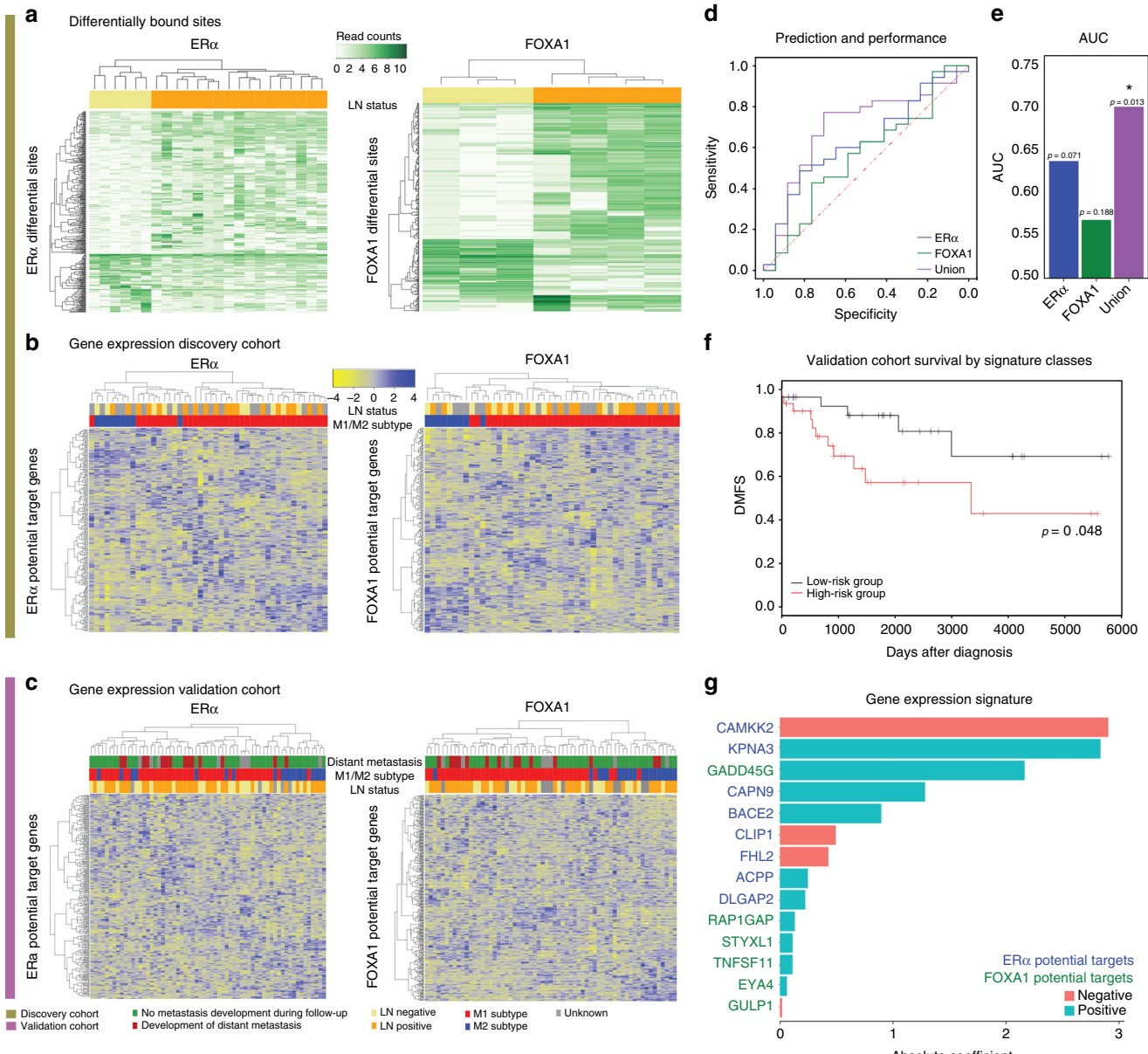

**Fig. 6** Genomic selectivity of ERα/FOXA1 action classifies male breast cancers on outcome. **a** Unsupervised hierarchical clustering of ERα and FOXA1 peak intensities at sites that classify on negative (yellow) or positive (orange) LN-status. **b** Unsupervised hierarchical clustering of the potential target genes driven from ERα and FOXA1 binding sites that classify on LN-status (discovery cohort). Negative (yellow) and positive (orange) LN-status, M1 (red), and M2 (blue) subtypes are indicated by the color bars above. **c** The same analysis as in **b**, using the validation cohort. Negative (yellow) and positive (orange) LN-status, M1 (red) and M2 (blue) subtypes as well as development of distant metastasis (red and green) are indicated by the color bars above. **d** Receiver-operator characteristic (ROC) curve indicating predictive performance of binary classification models trained with potential target genes of ERα (blue), FOXA1 (green) or the union of the two (purple). **e** Area under curve of the ROC curves of **d**, with *p*-values from bootstrapping analysis indicated on the top of each bar. **f** Kaplan–Meier analysis of two groups of patients (high- and low-risk groups in red and black lines, respectively) driven from the classification model trained with union of potential targets. Difference in survival is assessed with log rank test (*p*-value = 0.048). **g** Bar plot indicating 14 genes contributing to the classification model and their coefficients in the model. Predicted upstream regulator (blue − ERα and green − FOXA1) and sign of the coefficients (red − negative, green − positive) are indicated

Hills) followed by the UltraView Universal DAB Detection Kit (Ventana Medical Systems). GR was detected using clone D6H2L (1/600 dilution, 1 h at 37 °C, Cell Signaling) and visualized using Anti-Rabbit HQ (Ventana Medical systems) for 12 min at 37 °C followed by Anti-HQ HRP (Ventana Medical systems) for 12 min at 37 °C, followed by the ChromoMap DAB detection kit (Ventana Medical Systems). GATA3 was visualized using the OptiView DAB Detection Kit (Ventana Medical Systems). Slides were counterstained with Hematoxylin II and Bluing Reagent (Ventana Medical Systems). In the non-clinical setting (GR, FOXA1, GATA3), all scoring was performed on whole slides by a single pathologist (PJvD), blinded to patient status. In the clinical setting (ER, PR, and AR, we obtained

positive/negative status from the clinical records. In accordance with clinical guidelines in the United States[59], all samples were considered positive when at least 1% of nuclei were stained.

**ChIP-seq bioinformatics.** Sequenced samples were aligned to the reference human genome (Ensembl 37) using Burrows-Wheeler Aligner (BWA, v0.7.5a) with a mapping quality >20. Peak calling was performed using MACS (v1.4)[60] and DFilter (v1.5)[61], where only peaks were considered that were shared by the two peak callers. Genome browser snapshots were generated using IGV (v2.3.67), heatmaps were compiled using SeqMiner (v1.3.3)[62] and genomic region

enrichment analysis was performed with CEAS (v1.0.2)[63]. To generate consensus peaksets for each factor we used DiffBind (v1.14)[10] with a cutoff defined where a peak must be seen in at least 50% of the samples for that factor. In the event that there were only 2 samples sequenced for a factor, only peaks were considered found in both samples. A list of the union of these sites was generated for Fig. 2a, b). For two-way Venn diagrams, we used these cutoffs in each factor in the set of 4 (ERα vs AR), 3 (ERα vs FOXA1), and 2 (ERα vs GR/PR) (Fig. 2c, d). For differential binding analyses (Fig. 2e), DiffBind was used[10] with the following parameters, FDR of 0.1 for comparison between two different factors and p-value of 0.01 for comparison between different LN-status. Given a set of binding site, genomic distribution and significantly enriched motifs were obtained using CEAS[63] and SeqPos[64] in Galaxy Cistrome (v1.0.0)[64]. All identified motifs were included in the wordcloud figures without removing close homologues, to prevent selection bias (Supplementary Figures 5, 10 and 11).

**RNA isolation and RNA-seq.** Sections (30 μm thick) were cut from the frozen tumor tissues for RNA isolation. Total RNA was extracted using the *mir*Vana miRNA isolation kit (Ambion, USA) according to the manufacturer's protocol until the end of F1. Quality and quantity of the total RNA was assessed by the 2100 Bioanalyzer (Agilent, USA). Total RNA samples having RIN >8 were subjected to library generation.

Strand-specific libraries were generated using the TruSeq Stranded mRNA sample preparation kit (Illumina, USA; RS-122-2101/2) according to the manufacturer's instructions (Illumina, Part # 15031047 Rev. E). 3′ adenylated and adapter ligated cDNA fragments were subject to 12 cycles of PCR. The libraries were analyzed on a 2100 Bioanalyzer using a 7500 chip (Agilent, Santa Clara, CA), diluted and pooled equimolarly into a multiplexed, 10 nM sequencing pools and stored at −20 °C. Strand-specific cDNA libraries were sequenced with 100 base paired-end reads on a HiSeq2500 using V4 chemistry (Illumina).

**RNA-seq analysis.** For all analyses we used the reference file Ensembl GRCh37.75. Adapter filtered reads were subject to STAR alignment (v2.4.2)[65] carried out using default parameters. For expression analysis, HTSeq (v0.6.1p1)[66] was used to count reads for all genes in our RNA-seq samples using the htseq-count command. The DESeq2 (v1.16.0) R package was then used to generate a gene expression matrix from these data[67]. Normalization of the data was carried out using the 'rlog' method within the package. Only samples with at least 5 reads across each gene were retained for further analyses (N = 46).

**Prediction of male patient outcome on profiles of ERα and FOXA1.** We constructed a k-nearest neighbor classifier based on ERα binding profile of female patients using scikit-learn module (v0.19)[68] in Python. Taking read counts of ERα in the ERα binding sites that classify female outcome, male patient outcome is predicted by the outcome of five closest female data in terms of Minkowski distance.

**Logistic regression with elastic net regularization.** We used R package glmnet (v2.0)[43] for constructing a logistic regression model using RNA-seq data in our cohort. Leave-one-out cross validation was performed for finding robust coefficients. Taking the independent validation cohort, linear combination of gene expression levels using the coefficients (gene expression signature) were obtained to rank the patients. Measuring sensitivity and specificity of LN-status prediction with varying threshold gives a receiver-operating characteristics (ROC) curve from which area under the curve (AUC) can be measured. A threshold dividing high- and low-risk group was chosen as the median. We used potential target genes of ERα and FOXA1, and union of potential targets to construct three classification models. Note that due to the high dimensionality, no contributing gene is found when the model is trained with whole genome data.

**Bootstrapping analysis of classification model.** We used the R package boot (v1.3)[69] for bootstrapping analysis. For each classification model, 1000 random models were constructed in the same manner but with random gene sets with the same size to obtain bootstrapped AUC distribution. Taking the bootstrapped AUC distribution as a null hypothesis distribution, significance of the performance was assessed by one-tailed test.

**Survival analysis.** We used R package Survival (v2.38)[70] for survival analysis. Given two patient groups based on data availability, the log-rank test was used for overall survival (OS) comparison in our cohort and distant metastasis free survival (DMFS) comparison in a validation cohort. Cox regression was used to assess association of LN-status and gene expression signature to survival. The available additional prognostic factors used in multivariate Cox regression were LN-status, endocrine therapy, chemotherapy, radiotherapy and age at diagnosis.

**Data availability.** All ChIP-seq data generated in the study are available on GEO repository: GSE104399 and RNA-seq data on GEO repository: GSE104730. All public data streams used in the study are listed in Supplementary Table 3.

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

## Acknowledgements

We thank all patients who contributed to this study. The authors would like to acknowledge all the members of hospitals that provided tumor material. Wilbert Zwart is supported by the Bas Mulder Award from the Dutch Cancer Society (KWF)/ALpe d'HuZes and a VIDI grant from the Netherlands Organization for Scientific Research (NWO). JWMM received funding through the Cancer Genomics Netherlands from The Netherlands Organization for Scientific research (NWO). The authors would also like to acknowledge the effort and support of The Netherlands Cancer Institute (NKI) Genomics Core Facility. In addition, we would like to acknowledge the NKI Core Facility Molecular Pathology and Biobanking (CFMPB) for supplying Biobank material and lab support, especially Ingrid Hofland for GR IHC optimization.

## Author contributions

Experiments were performed by K.S., S.E.P.J., and W.Z. Computational analyses performed by T.M.S. and Y.K. Study supervision performed by P.v.d.G., P.J.v.D., S.C.L., L.W., and W.Z. Materials were provided by P.v.d.G., Q.F.M., J.W.M., C.H.M.v.D., E.B.,. I. H., P.B., V.T.H.B.M.S., and P.J.v.D., and processed by N.D.t.H., C.B.M., and K.S. Manuscript was written by T.M.S., Y.K., and W.Z. with input from all other authors.

## Additional information

**Competing interests:** The authors declare no competing financial interests.

