## [Peer Review File · Nature Communications]

Reviewers' comments:

Reviewer #1 (Remarks to the Author):

The work is guided by a straightforward genomics and bioinformatics analyses for ER positive male breast cancer (MBC) samples to characterise the molecular makeup of this disease. Contrary to what is known about ER/FOXA1 behaviour in female breast cancer cell lines and tissue, the authors of this study have observed FOXA1 profiles that do not correlate with ER binding profile. Moreover, a strong similarity in genomic profiles in ER and AR was found. The authors conclude that a cross talk between ER and AR might exist in male breast cancer and that ER might play a FOXA1-independent role. Although overall the data are well presented most of the findings in this study are not novel. Another major shortcoming is that all the data presented are just correlative. These findings might be true but difficult to believe without functional validation experiments.

The authors analysed ER and FOXA1 binding sites in MBC for outcome prediction. An unsupervised hierarchical clustering analysis of genes with differential ER and FOXA1 sites was performed. The analysis revealed clusters dominated by either M1 or M2 subtypes. The authors also state that global ER and FOXA1 chromatin binding selectivity reveals gender-specific prognostic features that classify MBC patients on survival. The conclusions derived from this study do not differ substantially from the study of Ross-Innes et al (PMID: 22217937). It is also difficult to see how this study contributes to improve the understanding of the tumour biology in male breast cancers. Moreover, this work does not establish whether these prognosis genes are functionally responsible for poor outcomes or simply markers of this phenotype. For instance, the data shown in figure 3 is just correlative and does not allow the authors to state that ER function may drive M1/M2 subtypes. Additional experiments in vitro are needed to clarify whether these genes are regulated by oestrogen and/or antagonised by anti-oestrogen receptor drugs.

I believe that the study is in a preliminary stage and the use of patient derived xenograft from male breast cancer samples might help the authors to validate their findings. Overall, the manuscript does not appear to provide sufficient new information that is of high novelty to the readership of Nature Communications.

Reviewer #2 (Remarks to the Author):

Manuscript title: The Epigenetic Regulation of Steroid Hormone Receptor Action in Male Breast Cancer.

In this paper, the authors investigated the action of steroid hormonal receptors, including Estrogen Receptor alpha (ER α), Androgen Receptor (AR), Progesterone Receptor (PR) and Glucocorticoid Receptor (GR) in male breast cancer (MBC) cases using chromatin immunoprecipitation coupled with massively-parallel sequencing (ChIP-seq).

The study suggests a genomic cross talk of steroid hormone receptor action in MBC, in relation to patient outcome.

Overall, the results presented in this paper are of interest in the field, but their relevance is limited by major problems related to the small number of cases included in the analyses performed (i.e. as shown in figure 1A and 1C, GR and PR binding events were observed in only 2 tumours, FOXA1 and AR binding events were evident in 7 and 9 tumours, respectively). Further analyses in additional series of breast tumours (also including female breast cancers) are necessary to confirm the results. As the authors observed, to date the potential interplay between different steroid hormone receptors has never been examined in human breast tumours, thus additional studies are required in order to suggest gender-specific features.

Major comments:

-The statement that resistance to hormonal treatment is common should be more clearly explained as in fact it is not frequently observed in MBC.

-The discussion should be more focused on the results presented (i.e. the observation that AR and GR seem to show no unique binding regions while ER α , FOXA1 and PR seem to show selective sites and the finding that ER α and AR show clustering within patient rather than between factors should be discussed). Furthermore, the limitations of the study due to the small number of cases analysed should be clearly stated and the results should be interpreted with caution.

Reviewer #3 (Remarks to the Author):

This manuscript by Severson, T. M. et al. profiles the chromatin occupancy of four steroid hormone receptors (SHRs), one potential 'pioneer' factor, and one 'enhancer' histone modification in male breast cancer (MBC) to begin to understand at a molecular level what may drive this rare breast cancer (1% of total breast cancers). Specifically, the authors characterize 47 estrogen receptor alpha (ER α) expressing MBC patient samples by defining the cistromes of four SHRs (ER α ; androgen receptor, AR; glucocorticoid receptor, GR; and progesterone receptor, PR), FOXA1 as a known pioneer factor in female breast cancer, and H3K4me1 as a known histone mark enriched at enhancers, as well as transcriptomics (RNA-seq) of these patient tumors (Figure 1). Previous publications exist for profiling MBC gene expression (e.g., Johansson, I. et al. 2012 Breast Cancer Res), and thus the RNA-seq done in this manuscript is not a novel concept. The authors found largely overlapping cistromes of the other SHRs with ER α , with AR and GR showing the highest degree of overlapping binding sites. Importantly, the FOXA1 cistrome is distinct from ER α (at least 68% of ER α bound sites lack FOXA1) and ER α binding localized mainly to promoter regions (Figure 2), both of which are in contrast to previous reports analyzing female breast cancer cells (Hurtado, A. et al. 2011 Nat. Genet.). Based upon a prior gene expression classification of MBC into luminal M1 or M2 subtypes (Johansson, I. et al. 2012 Breast Cancer Res), the authors found ER α binding sites unique to M1 or M2 (Figure 3). The authors also reported similar ER α binding sites (>50%) in MBC versus female breast cancers, but that ER α binding sites predictive for worst outcome in female breast cancer did not also apply to MBC (Figure 4 and Supplemental Figure 7). Even though FOXA1 does not appear to act as the 'key' pioneer factor for the majority of ER α binding events in MBC (Figure 2), its binding profile may provide predictive patient outcome information, along with ER α (Figure 5).

Since no MBC cell lines or mouse models exist, it is unfortunately difficult to perform intervention studies on this disease to infer more mechanistic insight into how all the SHRs and FOXA1 function to possibly drive MBC. In other words, how can one infer without knockdown/knockout their relative importance to MBC growth and metastasis? Therefore, this study is largely descriptive and correlative. However, this manuscript could nicely serve as a great resource for future studies. Additionally, there is missing information about the patients and key figures that needs to be provided prior to publication. Overall, the manuscript is not acceptable as is and requires major revisions as highlighted below.

Major issues:

1. The authors need to provide more information about the MBC patient samples and make Figure 1A clearer. While utilizing patient samples is certainly a strength of this study, the lack of patient information and a confusing figure format needs to be addressed. For example, it may be more useful to present the patient data in table format, similar to Johansson, I. et al. 2012 Breast Cancer Res. If the authors choose to keep Figure 1A in a heatmap-like format, the figure should be reformatted to address concerns listed below and the figure legend and/or results should be expanded. This study raises a number of questions about the patients that were not addressed in the manuscript:

- a. The authors cite in their most recent review (Severson, T. M. and Zwart, W. 2017 Endocr Relat Cancer) that MBC is frequently positive for ER α (91-95%) and/or PR (80-81%). But what is the actual expression status of the four SHRs in each breast cancer patient sample? Do all patients express all SHRs studied? If not, what percentages of patients express which receptor(s)?
- b. When heatmaps of SHR binding or Venn diagrams of overlapping peaks are shown (e.g., Figures

1C, 1D, 2, etc.), it is not clear which ERα cistromes (and sometimes other cistromes) the authors are analyzing and the rationale for choosing particular datasets (or is it an average or compilation of multiple datasets?). Please clarify this for the reader.

c. Information on whether or not these patients underwent endocrine therapy should be included and discussed. It would be of interest if a particular treatment regime changed a particular SHR cistrome (for example, the SERM tamoxifen is often given to MBC patients and this SERM can affect the ERα cistrome in breast cancers as the authors previously published in Severson, T.M. et al. 2016 Oncotarget).

d. The rationale on why the authors chose to look at the four SHR cistromes in select patients is not clear. For example, why did the authors choose to look at GR and PR cistromes in the two patients selected? Did these patients have high expression of these receptors? What about selecting patients for ChIP-seq of ERα, AR, FOXA1, and H3K4me1?

e. The authors do not explain why there is missing information in their patient data set, which may be confusing to readers. For example, they have 24 patients for whom there is survival data, but for the remaining 23 patients, is the survival data unknown? Or did the patients die of non-cancer causes?

2. The authors report that the FOXA1 cistrome in MBC does not correlate with the SHR cistromes, including ERα (lines 152-154, Figure 2B). However, when stratifying patients according to survival status, the union of differential ERα and FOXA1 binding sites together were predictive (Figures 5D-E). Can the authors elaborate on this and provide an explanation?

3. Additionally, the authors mention GATA3 expression is associated with FOXA1 and GR expression in ERα positive breast cancers in the Introduction (line 101). GATA3 has been published to affect distribution of ERα genome-wide chromatin occupancy in the MCF-7 female breast cancer cell line (Theodorou V. et al. 2013 Genome Res.). Also, GATA3 mutations have been reported in at least 15% of MBC (Piscuoglio S. et al. 2016 Clin Cancer Res.), thereby highlighting its possible functional importance in MBC. As FOXA1 does not appear to be the 'key' pioneer factor for ERα in MBC, it would be nice if ChIP-seq of other potential pioneer factors could be done from the same MBC patients. Therefore, it is highly recommended that the authors include a cistrome for another potential pioneer factor, such as GATA3, in their analysis.

4. The authors need to soften their language about the novelty of their study. For example, they state on line 123 that "each SHR was profiled in human tissue for the first time (>2 samples/factor)." However, there have been many ChIP-seq datasets generated on these SHRs in human tissues. The authors should clarify "in male breast cancer patient samples" or something similar. On line 106, the authors state: "to date all these transcription factors have never been profiled in human breast tumors." Here, the authors again should be more specific in their language and clarify that while all four SHRs have not been profiled together in a single study, clearly individual ones have been profiled (e.g., many papers published by the Brown and Carroll laboratories).

5. As this study is based heavily upon ChIP-seq data, what controls have been done or already published indicating the ERα, AR, PR, GR, FOXA1, and H3K4me1 antibodies cited in the Methods are truly 'specific' for their intended antigens? Could some of the ChIP-seq signal arise from other protein detection?

Minor issues:

1. The authors should describe their methods more thoroughly for identifying motifs or change their motif nomenclature, as the motifs for AR, GR, and PR are indistinguishable (reviewed in Mangelsdorf, D.J. et al. 1995 Cell). Also, are ESR1 and ESR2 bound sequences really that different (as it is known that ERα and ERβ can activate ERE-containing genes)?

2. For Line 126: can the authors clarify which subtype of MBC (M1 versus M2) is associated with poor prognosis? Johansson, I. et al. 2012 Breast Cancer Res described the luminal M1 subtype as being more aggressive and associated with a worst prognosis compared to the luminal M2, but the

text in this manuscript suggests otherwise.

3. Supplemental Figure 4 is very confusing. Based on Supplemental Figure 4A, the authors conclude that "ERa selective sites exposed more 'open' chromatin structure" (line 175). However, the authors need to change their nomenclature, as they did not truly measure chromatin accessibility (H3K4me1 is not a marker for chromatin accessibility—if the authors wish to conclude this, they would need to support this claim with experimental data from MNase-, DNase-, FAIRE-, or ATAC-seq, for example).

4. There is a lack of validation of their sequencing data by qPCR (for both ChIP-seq and RNA-seq datasets).

5. Data for ChIP-seq and RNA-seq need to be deposited to a publically-available database such as GEO or ArrayExpress to allow others to use the data to inform their hypotheses.

6. Title should be re-worded. The manuscript does not really address the "Epigenetic Regulation" of SHRs, as only one histone mark (H3K4me1) was profiled and DNA methylation was not profiled. Maybe a better title would be "Characterizing steroid hormone receptor chromatin binding landscapes in male breast cancer"?

7. On lines 135-136, it is stated "ERa-regions are bound by all the factors". This observed phenomenon is a first for MBC, but maybe the authors could cite that it was previously known different transcription factors 'pile up' with ERa in female breast cancer cells (e.g., with RAR, GATA3, FOXA1, etc.- Liu, Z. et al. 2014 Cell). Also, the Discussion section lines 260-263 seem to suggest ERa, AR, GR, and PR bind regions of DNA together due to closely juxtaposed EREs and ARE/GRE/PREs. However, a 'tethering' model whereby ERa might recruit some of the other SHRs (or vice versa) should be mentioned here as an alternative, given prior publications like Liu, Z. et al. that argue for such action.

8. How might knowing the 14 genes in Figure 5G be used in the clinic? For use as potential biomarkers? Could any of these be targeted by small molecule inhibitors (e.g., CAMKK2)?

9. Other edits for better clarity:

a) On lines 60-61 and 245, please change "enhancer-selective" to "enhancer-enriched" as H3K4me1 is often found at highly active promoter regions as well. b) On line 82 please add "(such as tamoxifen)" after "endocrine therapies". c) On line 89, replace "form" with "from"; on line 227, please add "to" after "(Methods)"; and on line 254 replace "then" with "than". d) Sentence on lines 93 and 94 starting with "AR activation in breast cancer cells..." cited references 19-21, but these studies used LNCaP prostate cancer cells, so please re-word this sentence or add different references. In fact, references 19-21 could be combined with references 22-25 later on line 95. e) Please add "in female breast cancer" after "ERa bound regions" on line 133. f) On line 139, it is not clear that "enhancer regions" were bound but that introns and distal intergenic regions were. Please change wording as one does not truly know these regions are all enhancing transcription. g) In Figure 3A, please use different colors for M1 and M2 than blue and red that are also used in the expression level scale. Also similar concern for Figure 3G (with green and purple representing both ChIP-seq versus RNA-seq as well as insulin signaling versus immune related) and Figure 5B (with red/blue for M1/M2 subtype as well as using these two colors for the heatmap color scale). h) On line 197-199, AR signaling was not the only pathway displaying better enrichment from ChIP-seq versus RNA-seq and thus this sentence needs re-wording. i) Why does Figure 5F show a p-value of 0.048, yet the figure legend (line 542) reports 0.041? j) Why are references 48 and 51 the exact same thing? Please delete one of these.

RESPONSE LETTER

-- Reviewer 1 --

We would like to thank the referees for their critical reading of our manuscript and their insightful comments, that have helped us to further strengthen our work.

Please note there are changes to the figures and supplemental figures and respective references in the text.

1. Original Figure 1A has been moved to New Supplemental Figure 1A and New Supplementary Table 1 to provide more clinical details on the samples.
2. Addition of New Supplemental Figures 2 and 3, showing qPCR and antibody validations.
3. Original Supplemental Figure 1 has been moved to new Supplemental Figure 4.
4. Addition of New Supplemental Figure 5 to show ER/AR overlap in female tumor and MCF7 cells.
5. Original Supplemental Figure 2 has been moved to New Supplemental Figure 6.
6. Original Supplemental Figure 3 has been moved to New Supplemental Figure 7.
7. Original Figure 2, ER and FOXA1 portion of Panel F and all of Panels G-H have been moved to New Figure 3A-C.
8. Original Figure 2, H3K4me1 portion of Panel F has been moved to New Supplemental Figure 8A.
9. Original Figure 2, Panel I has been moved to New Supplemental Figure 8B.
10. Original Supplemental Figure 4 has been removed.
11. Original Supplemental Figure 5 has been moved to New Supplemental Figure 9.
12. Original Supplemental Figure 6 has been moved to New Supplemental Figure 10.
13. Addition of New Supplemental Figure 11 to show ER binding shared between genders.
14. Addition of New Supplemental Figure 12 to show FOXA1 binding shared between genders.
15. Addition of New Supplemental Figure 13 to show clustering of all bindings sites for all factors between genders and cell lines.
16. Addition of New Supplemental Figure 14 to show clustering analysis of AR, GATA3, GR and PR binding sites between genders and cell lines.
17. Original Supplemental Figure 7 has been moved to New Supplemental Figure 15.
18. Original Supplemental Figure 8 has been moved to New Supplemental Figure 16.
19. Original Supplemental Figure 9 has been moved to New Supplemental Figure 17.
20. Addition of New Supplementary Table 2 to show patient characteristics of female tumor samples analyzed.

Reviewer 1 Comment 1	The conclusions derived from this study do not differ substantially from the study of Ross-Innes et al (PMID: 22217937). It is also difficult to see
--

	how this study contributes to improve the understanding of the tumour biology in male breast cancers.
Author Response	We believe that our current report goes far beyond the Nature paper from Ross-Innes and colleagues, as the study cited by the reviewer was performed exclusively in female breast tumors and examined only ER binding in relation to outcome. Our study is the first report to date studying chromatin binding profiles of multiple hormone receptors in male breast cancers and is the first report that examines in any human tumor the genome-wide chromatin binding profiles of 4 steroid hormone receptors (ER, AR, PR, GR), two pioneer factors (FOXA1, GATA3) and a histone modification (H3Kme1). With this, we find the comprehensive nature of this work to exceed any other prior ChIP-seq based reports in human solid tumors. With this, we are the first to show that:  1) Substantial overlap of chromatin binding exists between all steroid hormone receptors we studied in male breast cancer specimens: ER, AR, PR and GR. Given this is a pioneering report on studying multiple steroid hormone receptors in solid human tumors, this finding is completely novel in nature. 2) Even though the vast majority of ER sites are shared between male and female breast cancer, binding sites associated with outcome are gender-specific. Since treatment strategy of male breast cancer is strongly based on the female disease, identifying gender-specific features based on selective genomic actions of ER action between males and females is completely novel and of clinical relevance. 3) Between the intrinsic subtypes of male breast cancer, all steroid hormone receptors along with their pioneer factors, strongly deviate in chromatin binding profiles. This is a unique feature of male breast cancer, not described to date for female cases. This may potentially be in part the case, since no prior reports in human breast cancers studied these large numbers of patients. 4) In stark contrast to cell line reports, we now show for the first time that ER that is devoid of FOXA1 binding (representing around 60% of all ER sites) is very strongly promoter enriched, while all cell lines reported to date illustrate ER binding almost exclusively at enhancers. We now also provide new data, that this is not only the case for male breast cancer, but also for female tumors. With this, our report effectively rejects the current strong dogma, and we illustrate that ER in tumors is not the enhancer-acting transcription factor that we always thought

	it was, exclusively based on cell line studies. 5) Chromatin binding profiles of ER and AR almost perfectly overlap in male breast cancer, and we now also provide information that this is also the case for female breast tumors and cell lines. 6) Furthermore, in an important contribution to the field, we determined a gene expression signature that is significantly associated with distant-metastasis-free-survival in MBC. Outcome prediction is understudied in MBC making this finding even more clinically relevant. 7) Finally, since this is the first report describing the chromatin binding repertoire of multiple steroid hormone receptors in male breast cancers, where we now also provide the first-ever reported data on FOXA1 and AR ChIP-seq along with ER in female breast cancers, this report presents an invaluable resource for the community. With this, we feel we have provided compelling evidence that our study provides completely novel, exciting and unexpected new insights in male and now also female breast cancer, and we hope the reviewer agrees with us in this.
Reviewer 1 Comment 2	Moreover, this work does not establish whether these prognosis genes are functionally responsible for poor outcomes or simply markers of this phenotype. For instance, the data shown in figure 3 is just correlative and does not allow the authors to state that ER function may drive M1/M2 subtypes.
Author Response	Indeed, studies with clinical samples are correlative in nature, and mechanistic experiments are consequently not realistic without the availability of a model system that can be manipulated. In our study, we have established a gene signature for outcome prediction for which the functionality of individual genes can't be assessed; rather it is the combined effect of the genes that make up the association with poor outcome. To our knowledge, no examples in literature can be found in which a gene signature was examined for functional mechanistic relevance in tumorigenesis, using clinical specimens. We propose this signature is a biomarker for outcome prediction, not that it is responsible for outcome. The genes represented in Original Figure 3 are not the same genes as found in the gene signature as these are different analyses. In Original Figure 3 we aimed to compare and contrast the gene expression based M1/M2 genes (female breast cancer) and the genes associated with ER binding sites in M1/M2 male breast cancer samples. Through pathway

Manuscript changes	analysis, we identified both are associated with ER pathway, which suggests ER is associated with the gene expression based M1/M2 subtypes. Nonetheless, we do agree with the reviewer that it cannot be stated that ER function is driving the subtypes, since causality cannot be claimed using clinical specimens. We have included this statement in the discussion section. Line 305 added: 'While these data suggest ERα function may be driving these subtypes, causality can only be illustrated when cell line models, organoids or patient-derived xenografts are available for mechanistic studies.'
Reviewer 1 Comment 3	Additional experiments in vitro are needed to clarify whether these genes are regulated by oestrogen and/or antagonised by anti-oestrogen receptor drugs. I believe that the study is in a preliminary stage and the use of patient derived xenograft from male breast cancer samples might help the authors to validate their findings. Overall, the manuscript does not appear to provide sufficient new information that is of high novelty to the readership of Nature Communications.
Author Response	Even though we agree with the reviewer that it would be of interest to study a patient-derived xenograft for male breast cancer, we respectfully find this request unrealistic and quite impossible to address, as in vitro models for male breast cancer do not exist. Patient derived xenografts are known to have an extremely low engraftment rate in ER positive female breast cancers of 1-2% (DeRose et al., Nat Med 2011;17:1514-20 and Zhang et al., Cancer Res 2013;73:4885-97) making the use of PDX models both financially and technically unfeasible. Furthermore, since male breast cancer is a rare disease, PDX models of male breast cancer are very unlikely to be generated by anyone in the near future. Furthermore, we strongly disagree with the reviewer, stating the manuscript provides limited new information, as extensively addressed in our response to comment 1.

-- Reviewer 2 --

We would like to thank the referees for their critical reading of our manuscript and their insightful comments, that have helped us to further strengthen our work.

Please note there are changes to the figures and supplemental figures and respective references in the text.

1. Original Figure 1A has been moved to New Supplemental Figure 1A and New Supplementary Table 1 to provide more clinical details on the samples.
2. Addition of New Supplemental Figures 2 and 3, showing qPCR and antibody validations.
3. Original Supplemental Figure 1 has been moved to new Supplemental Figure 4.
4. Addition of New Supplemental Figure 5 to show ER/AR overlap in female tumor and MCF7 cells.
5. Original Supplemental Figure 2 has been moved to New Supplemental Figure 6.
6. Original Supplemental Figure 3 has been moved to New Supplemental Figure 7.
7. Original Figure 2, ER and FOXA1 portion of Panel F and all of Panels G-H have been moved to New Figure 3A-C.
8. Original Figure 2, H3K4me1 portion of Panel F has been moved to New Supplemental Figure 8A.
9. Original Figure 2, Panel I has been moved to New Supplemental Figure 8B.
10. Original Supplemental Figure 4 has been removed.
11. Original Supplemental Figure 5 has been moved to New Supplemental Figure 9.
12. Original Supplemental Figure 6 has been moved to New Supplemental Figure 10.
13. Addition of New Supplemental Figure 11 to show ER binding shared between genders.
14. Addition of New Supplemental Figure 12 to show FOXA1 binding shared between genders.
15. Addition of New Supplemental Figure 13 to show clustering of all bindings sites for all factors between genders and cell lines.
16. Addition of New Supplemental Figure 14 to show clustering analysis of AR, GATA3, GR and PR binding sites between genders and cell lines.
17. Original Supplemental Figure 7 has been moved to New Supplemental Figure 15.
18. Original Supplemental Figure 8 has been moved to New Supplemental Figure 16.
19. Original Supplemental Figure 9 has been moved to New Supplemental Figure 17.
20. Addition of New Supplementary Table 2 to show patient characteristics of female tumor samples analyzed.

Reviewer 2 Comment 1	Overall, the results presented in this paper are of interest in the field, but their relevance is limited by major problems related to the small number of cases included in the analyses performed (i.e. as shown in
---

	figure 1A and 1C, GR and PR binding events were observed in only 2 tumours, FOXA1 and AR binding events were evident in 7 and 9 tumours, respectively). Further analyses in additional series of breast tumours (also including female breast cancers) are necessary to confirm the results. As the authors observed, to date the potential interplay between different steroid hormone receptors has never been examined in human breast tumours, thus additional studies are required in order to suggest gender-specific features.
Author Response	We are delighted to hear the reviewer finds our work of interest to the field, and (even though we do provide the largest ChIP-seq study to date on solid tumors) we agree that additional datasets were required to further confirm the results. Therefore, we have substantially increased the number of ChIP-seq samples used for analyses, and provide now a total of 161 ChIP-seq datasets of with 110 from human tumors. For male breast cancers, we have added an additional 13 ChIP-seq datasets (total number = 67), including 2 ChIP-seq samples for ER, 1 for AR, 5 for GR and 2 for PR. With these additional data, we are able to more reliably determine common binding events between factors. Especially for GR and PR, the data are now significantly stronger, yet without having any effects on the original conclusions from the previous submission. In addition, to address another reviewer's question about GATA3, we added ChIP-seq for 3 GATA3 samples for the series. As this reviewer suggests, for a more thorough comparison to female tumors, we generated 6 ER, 7 FOXA1 and 1 AR ChIP-seq data from 8 female tumors. Further, we included 12 ER ChIP-seq datasets from Jansen et. al to increase the number of female clinical tumor samples studied. We also put our findings in the context of well-known female breast cancer cell lines, by including 51 ChIP-seq runs from MCF7, ZR751, T47D and xenografts derived from MCF7 cells. With the addition of these new datasets, we corroborated our original findings 1) that FOXA1 binding sites are separated from the other factors, including ER and 2) that there is a significant portion of ER sites which are not enriched for FOXA1. Importantly, both for male and female breast tumors, we find that ER sites devoid of FOXA1 binding are strongly promoter enriched, which is completely against the current cell line-based dogma on ER being an enhancer-selective transcription factor. These results are now included in the new Figure 4, in which we directly compared ER profiles (devoid of FOXA1) between male breast tumors, female breast tumors and cell lines. We hope that, with the inclusion of an additional 86 ChIP-seq datasets in the study, with a total number of 161, the reviewer finds our dataset

Manuscript changes	sufficiently increased. Furthermore, we would like to thank the reviewer for suggesting the additional validation of our findings in female breast tumors, which resulted in the observation of promoter-enriched ER sites also in females, making us believe the enhancer-specific effects of ER as is the current state-of-the-art may be the result of over-extrapolating cell line results. Throughout the text: Added GATA3 to all relevant datastreams and included in the text. Throughout the text: Added reference to female data and cell line where appropriate. Line 57: added: 'the female disease and'. Line 59 added: 'human breast tumors of both genders'. Line 103 added: 'FBC'. Line 116 added: 'which we compared with the female counterpart'. Line 128 added: 'and integrated these data with gene expression and compared with female breast cancer and cell line data.' Line 160 added: 'All sites shared for individual factors: ERα (15 out of 30 samples), AR (5 out of 10 samples), FOXA1 (3 out of 7 samples), PR (2 out of 4 samples), GR (3 out of 7 samples) and GATA3 (1 out of 3 samples).' Line 169 add: 'Contrary to what is described for ERα/FOXA1 behavior in FBC cell lines¹², we observed FOXA1 profiles do not correlate with other profiles, while all SHRs and GATA3 cluster together. Most notably, a strong similarity in genomic profiles in ERα and AR was found, in which practically all AR sites were co-occupied with ERα (Fig. 2C), which was also seen in female breast tumor and MCF7 cells (Supplementary Fig. 5A, B). ' Line 178 add: 'A significant proportion of ERα binding sites overlap with AR and GR sites as seen in cell line data^{44,45} which we confirmed in MBC (Fig. 2C-D). Interestingly, although PR modulates ERα binding in FBC³⁰, many PR sites in MBC were devoid of ERα (56%). These findings were
--

Figure changes	confirmed in DNA binding correlation matrices (Fig. 2E). ERα and AR show clustering within patient rather than between factors (Fig. 2E), indicating a stronger correlation between factors within the same tumor as compared to the same factor between tumors.' Line 188 add: 'In line with literature¹², we find ~50% overlap between ERα and FOXA1 in cell lines, where sites selectively enriched for FOXA1 or ERα (Fig. 3A, B and Supplementary Fig. S7) were largely found in intronic and distal intergenic regions (Figure 3C). These findings are in stark contrast to observations in both male and female breast cancers, in which ERα sites devoid of FOXA1 were strongly promoter-enriched (Fig. 3A-C), suggesting the model systems currently used do not adequately capture the genomic distributions of ERα found in clinical samples.' Line 194 add: 'In contrast to FOXA1-enriched sites, sites selectively occupied by ERα were weaker for active enhancer mark H3K4me1³⁸ (Supplementary Fig. 8A,B). Interestingly, GATA3 was found at both the FOXA1-enriched and ERα-enriched sites (Supplementary Fig.8A). Motifs at ERα selective sites are related to ESR1 and devoid of forkhead motifs (Supplementary Fig. 9), which is in contrast to the total of ERα sites (Supplementary Fig. 4).' Line 218 added: 'in line with the strong overlap of AR/ERα binding in these tumors (Fig. 2 and Fig. 4C).' Line 229 added: 'at comparable intensity'. Line 231 added; 'While clear clustering was observed for ERα between (male and female) tumors and cell lines (Fig. 5A), no separation on gender was observed for any of the factors studied in an integrative analysis (Supplementary Fig. 13) as well as separately for all factors studied (Fig. 5A, B, Supplementary Fig. 14).' Line 275 added: ' both male and female' Revised Figure 1-2, and Figure 5 (that includes multiple female data series and model system data). Figure 1A now shows the experimental and analytical set-up. Revised figure 2 now adds GATA3 data to each analysis. For clarity and space considerations, we have moved panels H-J of previous Figure 2 to a new Figure 3 and new Supplemental Figure 4 and 8.
---

Reviewer 2 Comment 2	The statement that resistance to hormonal treatment is common should be more clearly explained as in fact it is not frequently observed in MBC.
Author Response	Relapse after adjuvant hormonal therapy has been noted in male breast cancer (Ruddy et al. 2013, Ann Oncol., Korde et al. 2010, JCO, Cutuli et al. 2010, Critical Rev. in Onc/Hem. Abreu et al. Int. J. Cancer, 2016). However, to soften the statement, we will remove resistance as the cause of the relapse is unknown in most cases.
Manuscript change	Line 84 add: Change 'resistance is common' to 'relapse after hormonal treatment has been noted as in female breast cancer (FBC) ^{2,7,8} .
Reviewer 2 Comment 3	The discussion should be more focused on the results presented (i.e. the observation that AR and GR seem to show no unique binding regions while ER α , FOXA1 and PR seem to show selective sites and the finding that ER α and AR show clustering within patient rather than between factors should be discussed).
Author Response	We thank the reviewer for highlighting this issue, and we have modified the discussion accordingly. We would like to highlight that (as mentioned above to comment 1), we have sequenced many additional ChIP-seq samples for the male breast tumors (n=13), substantially increasing the number of samples studied for GR and PR. Even though the number of samples has increased, the conclusions however remained unaltered, in which we find very strong overlap between ER/AR and ER/GR, while this is less the case for ER/FOXA1 and ER/PR. In addition, the newly introduced female data also further strengthen these statements.
Manuscript changes	As suggested by the reviewer, the discussion now stays closer to the actual results and the description thereof. Line 281 added: 'Our data reveal that genomic functions of ER α and AR in male breast tumors are largely overlapping, which strongly co-localized with GR and PR at the same regions. Even though many sites for GATA3, FOXA1 and PR were not shared with ER α , both AR and GR show virtually no unique binding sites with respect to ER α binding. AR has been shown to compensate for ER α in ER α -/AR+ female breast cancers ^{45,49} , however the biological interaction between ER α and AR is relatively unknown in both FBC and MBC ⁵⁰ .'
Reviewer 2 Comment 4	Furthermore, the limitations of the study due to the small number of cases analysed should be clearly stated and the results should be

	interpreted with caution.
Author Response	We fully agree with the reviewer, and even though this is to our knowledge the largest ChIP-seq study to date on solid human tumors, in which we added 2 additional cases (MBC) and 13 additional ChIP-seq samples to strengthen our conclusions, and included additional data for female breast cancer and female breast cancer cell-lines to more adequately describe differences between genders. Nonetheless, the reviewer is absolutely right that for some factors we still have limited datastreams, Furthermore, as this is a rare cancer and we perform ChIP-seq on clinical samples, numbers will always be low. As suggested by the reviewer, we now included a statement in the discussion section, in which the limitations of the study due to the small number of analysed cases are stated and results should be treated with caution.
Manuscript changes	Line 297 added: 'As MBC is a rare cancer with limited numbers of available tumors for genomic studies, and ChIP-seq analyses for some factors including GR and PR were on relatively low number of tumors, these results should however be validated in larger cohorts.'

-- Reviewer 3 --

We would like to thank the referees for their critical reading of our manuscript and their insightful comments, that have helped us to further strengthen our work.

Please note there are changes to the figures and supplemental figures and respective references in the text.

1. Original Figure 1A has been moved to New Supplemental Figure 1A and New Supplementary Table 1 to provide more clinical details on the samples.
2. Addition of New Supplemental Figures 2 and 3, showing qPCR and antibody validations.
3. Original Supplemental Figure 1 has been moved to new Supplemental Figure 4.
4. Addition of New Supplemental Figure 5 to show ER/AR overlap in female tumor and MCF7 cells.
5. Original Supplemental Figure 2 has been moved to New Supplemental Figure 6.
6. Original Supplemental Figure 3 has been moved to New Supplemental Figure 7.
7. Original Figure 2, ER and FOXA1 portion of Panel F and all of Panels G-H have been moved to New Figure 3A-C.
8. Original Figure 2, H3K4me1 portion of Panel F has been moved to New Supplemental Figure 8A.
9. Original Figure 2, Panel I has been moved to New Supplemental Figure 8B.
10. Original Supplemental Figure 4 has been removed.
11. Original Supplemental Figure 5 has been moved to New Supplemental Figure 9.
12. Original Supplemental Figure 6 has been moved to New Supplemental Figure 10.
13. Addition of New Supplemental Figure 11 to show ER binding shared between genders.
14. Addition of New Supplemental Figure 12 to show FOXA1 binding shared between genders.
15. Addition of New Supplemental Figure 13 to show clustering of all bindings sites for all factors between genders and cell lines.
16. Addition of New Supplemental Figure 14 to show clustering analysis of AR, GATA3, GR and PR binding sites between genders and cell lines.
17. Original Supplemental Figure 7 has been moved to New Supplemental Figure 15.
18. Original Supplemental Figure 8 has been moved to New Supplemental Figure 16.
19. Original Supplemental Figure 9 has been moved to New Supplemental Figure 17.
20. Addition of New Supplementary Table 2 to show patient characteristics of female tumor samples analyzed.

Reviewer 3 Major issues comment 1	The authors need to provide more information about the MBC patient samples and make Figure 1A clearer. While utilizing patient samples is certainly a strength of this study, the lack of patient information and a
--	---

	confusing figure format needs to be addressed. For example, it may be more useful to present the patient data in table format, similar to Johansson, I. et al. 2012 Breast Cancer Res. If the authors choose to keep Figure 1A in a heatmap-like format, the figure should be reformatted to address concerns listed below and the figure legend and/or results should be expanded.
Author Response	We understand that some readers prefer a table format to show the clinical and molecular data collected more extensively. We have now replaced previous Figure 1A with Table 1, enabling us to be more clear on the patient information. Because the table does not give comprehensively overview of all information at the level of the individual we retain the original heatmap and moved it to Supplementary Figure 1A. Furthermore, we have replaced this figure with a schematic to show the study analysis process (New Figure 1A).
Manuscript changes	Line 127 after 'specimens' added: 'integrated these data with gene expression and compared with female breast cancer and cell line data (Fig. 1A, Supplementary Fig. 1A).' Added references to supplemental Figure 1A where appropriate. Line 141: added 'and Supplementary Table 1'. Line 154 changed Sup. Fig.4.
Figure changes	Previous Figure 1A is moved to new Supplementary Table1. New Figure 1A has been generated to show a schematic of the study. Original Supplemental Figure 1 is changed to new Supplemental Figure 4.
Reviewer 3 Major issues comment 1A	This study raises a number of questions about the patients that were not addressed in the manuscript: a. The authors cite in their most recent review (Severson, T. M. and Zwart, W. 2017 Endocr Relat Cancer) that MBC is frequently positive for ERa (91-95%) and/or PR (80-81%). But what is the actual expression status of the four SHRs in each breast cancer patient sample? Do all patients express all SHRs studied? If not, what percentages of patients express which receptor(s)?
Author Response	This is a very relevant issue, and we thank the reviewer for bringing this to our attention. To assess this, we obtained staining data or stained all male breast cancer FFPE blocks available on IHC for ER, AR, PR, GR,

Manuscript changes	GATA3 and FOXA1. All analyses tumors were positive for all factors we studied, and this information has now been included in Supplementary Table 1. These data have been added to new Supplementary Table 1.
Reviewer 3 Major issues comment 1B	b. When heatmaps of SHR binding or Venn diagrams of overlapping peaks are shown (e.g., Figures 1C, 1D, 2, etc.), it is not clear which ERα cistromes (and sometimes other cistromes) the authors are analyzing and the rationale for choosing particular datasets (or is it an average or compilation of multiple datasets?). Please clarify this for the reader.
Author Response Manuscript changes	We apologise for the confusion and now realize that we did not adequately describe our methodologies for some analyses. For all analyses in which we studied SHR individually (without looking at overlap with other factors, so for Figure 1 we analysed all peaks identified in 50% of the samples. All analyses in which two factors were compared (Figure 2C-E) pairwise analyses were performed. For example in analysing the ER/FOXA1 overlap (Figure 2C), we only assessed ER data from the same tumor of which we also had FOXA1 data available. We believe this pairwise analyses are intrinsically cleaner and more reliable, but this does result in different number of ER sites studied for each comparison. We have included this information and the explanation thereof in the results section, methods section and figure legends. We have added the criteria for defining consensus in the legend of Figure 1C Figure 2C. Line 158 added: ‘We have shown ERα binding sites have considerable overlap with other factors studied. Next, overlap of ERα with the other steroid hormone receptors (Fig. 2A, left) or pioneer factors FOXA1 and GATA3 (Fig. 2A, right) was studied. All sites shared for individual factors: ERα (15 out of 30 samples), AR (5 out of 10 samples), FOXA1 (3 out of 7 samples), PR (2 out of 4 samples), GR (3 out of 7 samples) and GATA3 (1 out of 3 samples). AR and GR have virtually no unique binding regions, while selective sites for ERα and PR are identified. When examining ERα, FOXA1 and GATA3, we identified selective sites in each. Strikingly, 79% of FOXA1 sites were FOXA1-selective while ER and GATA3-selective sites were 42% and 33% respectively. Next, we counted the reads in the union of sites for all factors (Fig. 2A) and

Manuscript changes, cont.	computed the correlation score for each sample (Pearson correlation coefficient), which was represented in a correlation matrix (Fig. 2B).’ Legend for Figure 1C: Heatmaps depicting peak intensity in primary tumors for 30 ERα (blue), 7 FOXA1 (light green), 10 AR (red), 7 GR (black), 4 PR (purple) and 3 GATA3 (dark green) binding events (\pm 5kb from the peak center (triangle)). 5079 ERα sites were determined using the consensus ERα binding sites identified in at least 50% of patients (15 out of 30 samples). Legend for Figure 2C: Individual Venn diagrams depicting the pairwise overlap of ERα with each factor. Consensus binding sites for each factor were used (Fig. 1D), and overlap with ERα sites from the same tumors was assessed, taking the same threshold.
Reviewer 3 Major issues comment 1C	c. Information on whether or not these patients underwent endocrine therapy should be included and discussed. It would be of interest if a particular treatment regime changed a particular SHR cistrome (for example, the SERM tamoxifen is often given to MBC patients and this SERM can affect the ERα cistrome in breast cancers as the authors previously published in Severson, T.M. et al. 2016 Oncotarget).
Author Response Manuscript changes	We would like to highlight that all the breast cancers reported in this study are from treatment-naïve patients prior to surgery. The paper the reviewer refers to is of female breast cancer patients who received neoadjuvant treatment, thus prior to surgery. Neoadjuvant endocrine therapy is not a standard procedure in clinical practice, and the very vast majority of all patients exclusively receive endocrine therapeutics after surgery. This information has now been more explicitly included in the paper Nonetheless, we fully agree that information on adjuvant treatments are of interest to the reader and as such we have collected all available information regarding surgery, chemotherapy, radiotherapy and added it to new Table 1. A new Supplementary table 1 containing these data has been added to the manuscript. Line 127 added; ‘ from patients who did not receive any therapy prior to surgery’
Reviewer 3 Major issues	d. The rationale on why the authors chose to look at the four SHR cistromes in select patients is not clear. For example, why did the

comment 1D	authors choose to look at GR and PR cistromes in the two patients selected? Did these patients have high expression of these receptors? What about selecting patients for CHIP-seq of ERa, AR, FOXA1, and H3K4me1?
Author Response	To avoid a selection bias, we first attempted ER ChIP-seq on all samples. For all samples with ER ChIP-seq, we randomly selected samples with succesful ER ChIP for FOXA1 (as it is the pioneering factor for ER in FBC) and AR. With considerable new interest in the field for additional SHRs and their interplay with ER (ER/PR: Mohammed., Nature and ER/GR: Miranda et al Cancer Research 2013, Covro et al J Immunol, 2011, Bolt et al NAR 2013), GR and PR were included at a later stage, for which samples were randomly selected, as goes for H3K4me1. As suggested by review #3, we also added CHIP-seq for GATA3 for three randomly selected tumors. As such, the selection of samples to be ChIPped for a specific factor was quite random and no prior selection based on expression levels on any other clinicopathological parameters was made.
Manuscript changes	In order to show that we did not select samples specifically based on expression of these receptors, we have added these data to a new Supplementary Table 1. Line 130 added: 'Samples were selected randomly for ChIP-seq of different factors.'
Reviewer 3 Major issues comment 1E	e. The authors do not explain why there is missing information in their patient data set, which may be confusing to readers. For example, they have 24 patients for whom there is survival data, but for the remaining 23 patients, is the survival data unknown? Or did the patients die of non-cancer causes?
Author Response	Unfortunately as this was a retrospective series on a rare cancer, we were unable to retrieve all relevant clinical data for each patient. Of patients where we indicate there is no survival data, the survival data are unknown. In Supplemental figure 16, we show that LN status of patient at diagnosis can highly significantly predict the overall survival, which makes us believe that most of these patients died from cancer.

Manuscript changes	Missing clinical data are now clearly indicated and described in new Supplementary Table 1. Line 142 added: 'Missing clinical data are indicated in Supplementary Table 1'.
Reviewer 3 Major issues comment 2	2. The authors report that the FOXA1 cistrome in MBC does not correlate with the SHR cistromes, including ERa (lines 152-154, Figure 2B). However, when stratifying patients according to survival status, the union of differential ERa and FOXA1 binding sites together were predictive (Figures 5D-E). Can the authors elaborate on this and provide an explanation?
Author Response	Indeed, we reveal that in male breast cancers, although overlap between ER and FOXA1 is found, many FOXA1 sites we identified were devoid of ER signal. Since the union of genes with differential ER and FOXA1 binding outperformed the individual classifications, we hypothesize that an additional transcription factor which binds the FOXA1-enriched sites may drive tumor progression, together with the above-mentioned ER action. We tested whether the other SHRs we studied were found at the FOXA1-enriched regions, but this was not the case. With that, although we provide evidence that the union of genes under differential control of ER and FOXA1 jointly classify patients on outcome, it remains to be determined which transcription factor is facilitated by FOXA1 at these sites. This argumentation is now also incorporated in the discussion section.
Manuscript changes	Line 311 added: 'The union of genes under differential control of ER and FOXA1 jointly classify patients on outcome, and it remains to be determined which transcription factor is facilitated by FOXA1 at these sites.'
Figure Changes	Figure 3 (new) will now shows ER vs FOXA1 in cell line, female and male. In each biological system, we identified a substantial amount of ER sites are devoid of FOXA1.
Reviewer 3 Major issues comment 3	3. Additionally, the authors mention GATA3 expression is associated with FOXA1 and GR expression in ERa positive breast cancers in the Introduction (line 101). GATA3 has been published to affect distribution of ERa genome-wide chromatin occupancy in the MCF-7 female breast cancer cell line (Theodorou V. et al. 2013 Genome Res.). Also, GATA3 mutations have been reported in at least 15% of MBC (Piscuoglio S. et al. 2016 Clin Cancer Res.), thereby highlighting its possible functional importance in MBC. As FOXA1 does not appear to

	be the 'key' pioneer factor for ERα in MBC, it would be nice if CHIP-seq of other potential pioneer factors could be done from the same MBC patients. Therefore, it is highly recommended that the authors include a cistrome for another potential pioneer factor, such as GATA3, in their analysis.
Author Response Manuscript changes	We thank the reviewer for this excellent suggestion. We have now performed GATA3 ChIP-seq analysis for 3 tumors and found in male breast cancer that there are factor specific sites for ER, FOXA and GATA3. We show in cluster analysis that GATA3 does not cluster separately from ER, but FOXA1 still maintains its own space in the cluster (Figure 2B). Furthermore, when focussing on the ER sites devoid of FOXA1 (Supplemental Figure 8A) GATA3 was found associated at these sites. Cummulatively, these findings suggest it is possible that GATA3 is taking over as a key pioneer factor in MBC. Throughout the text: Added GATA3 to all relevant datastreams and included in the text. Line 159 to 166 has been changed to read: 'Next, overlap of ERα with the other steroid hormone receptors (Fig. 2A, left) or pioneer factors FOXA1 and GATA3 (Fig. 2A, right) was studied. All sites shared for individual factors: ERα (15 out of 30 samples), AR (5 out of 10 samples), FOXA1 (3 out of 7 samples), PR (2 out of 4 samples), GR (3 out of 7 samples) and GATA3 (1 out of 3 samples). AR and GR have virtually no unique binding regions, while selective sites for ERα and PR are identified. When examining ERα, FOXA1 and GATA3, we identified selective sites in each. Strikingly, 79% of FOXA1 sites were FOXA1-selective while ER and GATA3-selective sites were 42% and 33% respectively.' Line 194, added: 'In contrast to FOXA1-enriched sites, sites selectively occupied by ERα were weaker for active enhancer mark H3K4me1³⁸ (Supplementary Fig. 8A,B). Interestingly, GATA3 was found at both the FOXA1-enriched and ERα-enriched sites (Supplementary Fig.8A). Motifs at ERα selective sites are related to ESR1 and devoid of forkhead motifs (Supplementary Fig. 9), which is in contrast to the total of ERα sites (Supplementary Fig. 4).'
Figure Changes	The original Venn diagram in Figure 2A has been changed to two Venn diagrams, one showing only SHRs and one showing ER/FOXA1 and

Figure Changes, cont.	GATA3. Other figures have been updated to include GATA3 data, Figure 1B-E, Figure 2, and Supplementary Figure 4, Supplementary Figure 6 and Supplementary Figure 8A.
Reviewer 3 Major issues comment 4	4. The authors need to soften their language about the novelty of their study. For example, they state on line 123 that “each SHR was profiled in human tissue for the first time (>2 samples/factor).” However, there have been many ChIP-seq datasets generated on these SHRs in human tissues. The authors should clarify “in male breast cancer patient samples” or something similar. On line 106, the authors state: “to date all these transcription factors have never been profiled in human breast tumors.” Here, the authors again should be more specific in their language and clarify that while all four SHRs have not been profiled together in a single study, clearly individual ones have been profiled (e.g., many papers published by the Brown and Carroll laboratories).
Author Response Manuscript changes	We apologise for this, and by no means wanted to imply that we are the first who performed ChIP-seq for ER and AR in tumors. We have adjusted the text to better appreciate the seminal work from others. To our knowledge, this is however the first report in which all these factors were profiled in the same tumor samples. Furthermore, we would like to highlight that to our knowledge this is also the first report in which human breast tumor specimens were ChIPped for AR, GR, PR, FOXA1 and H3K4me1. We have updated the text accordingly. Line 107: added ‘Although ERα cistromics has previously been studied in female breast tumors^{10,11}, and its interplay with transcription factors has been reported in cell lines^{11,12,14,32-37}, all these transcription factors have never been profiled together in a single study in human breast tumors.’ Line 131: added “MBC for the first time” before “(>= 3 samples/factor)”. Line 110: added “ together in a single study” after “never been profiled”.
Reviewer 3 Major issues comment 5	5. As this study is based heavily upon ChIP-seq data, what controls have been done or already published indicating the ERα, AR, PR, GR, FOXA1, and H3K4me1 antibodies cited in the Methods are truly ‘specific’ for their intended antigens? Could some of the ChIP-seq signal arise from other protein detection?

Author Response	To determine antibody specificity for ER, AR, PR and GR we transiently overexpressed all SHRs of interest individually in U2OS cells, which are devoid of all factors we studied. Next, we stained all slides with the same antibodies for which ChIP-seq was performed, exclusively showing signal when the corresponding SHR was overexpressed (Supplemental Figure 3A). For the antibody used for FOXA1 ChIP-seq, we already reported previously that it detects both FOXA1 as well as FOXA2 (Droog et al., 2016 Cancer Research). However, since FOXA2 is not expressed in the tumor samples while high FOXA1 levels are detected (new supplementart Figure 3B), this cross-reactivity is not a matter of concern in our samples. The ab8895 antibody for H3K4me1 is highly specific for this histone modification, as has been validated by others. http://www.histoneantibodies.com/FinalArrayData/H3K4me1/ http://compbio.med.harvard.edu/antibodies/antibodies/72 Line 145, add: ‘using validated antibodies (see Methods, Sup. Fig. 3A, 3B)’ Line 375: added to Methods section: ‘SHR antibody specificity was validated by immunohistochemistry on U2Os cells transiently transfected with any of the SHRs of interest (Supplementary Fig. 3A). Antibody used for FOXA1 ChiP-seq also detects FOXA2⁵⁶ but only FOXA1 is expressed in these tumors. (Supplementary Fig 3B). Specificity for the antibody used for H3K4me1 ChiP-seq (ab8895) has been validated by others^{57,58}.’ New Supplementary Figure 2 and new Supplementary Figure 3.
Manuscript changes	
Figure changes	
Reviewer 3 Minor issues comment 1	The authors should describe their methods more thoroughly for identifying motifs or change their motif nomenclature, as the motifs for AR, GR, and PR are indistinguishable (reviewed in Mangelsdorf, D.J. et al. 1995 Cell). Also, are ESR1 and ESR2 bound sequences really that different (as it is known that ERa and ERbeta can activate ERE-containing genes)?
Author Response	We used SeqPos (He et al. Nat. Gen 2010) which is a routine methodology for identifying sequence motifs in a given region. This method checks for sequence similarity as well as verifying that the motif occurs at the center of the region (summit of the peak). An intrinsic problem with motif analysis in steroid hormone receptor biology is that indeed the motifs are very similar. We chose to keep all motifs in our wordclouds (Supplemental Figures 4, 9 and 10) (for

Manuscript changes Figure changes	example, not removing ESR2 in ERα binding regions) because removing resembling motifs may introduce a bias. Line 419: added ‘All identified motifs were included in the wordcloud figures without removing close homologues, to prevent selection bias (Supplemental Figures 4, 9 and 10).’ We have changed original Supplementary Figure 1 to new Supplementary Figure 4.
Reviewer 3 Minor issues comment 12	For Line 126: can the authors clarify which subtype of MBC (M1 versus M2) is associated with poor prognosis? Johansson, I. et al. 2012 Breast Cancer Res described the luminal M1 subtype as being more aggressive and associated with a worst prognosis compared to the luminal M2, but the text in this manuscript suggests otherwise.
Author Response Manuscript changes	We thank the reviewer for noticing this mistake in the text. We have updated the text to reflect the Johansson definitions. Please note that throughout the figures and analyses, the correct annotations were used. Line 134 change: ‘M1(good) and M2(poor)’ to M1 (poor) and M2 (good)’
Reviewer 3 Minor issues comment 3	Supplemental Figure 4 is very confusing. Based on Supplemental Figure 4A, the authors conclude that “ERα selective sites exposed more ‘open’ chromatin structure” (line 175). However, the authors need to change their nomenclature, as they did not truly measure chromatin accessibility (H3K4me1 is not a marker for chromatin accessibility-if the authors wish to conclude this, they would need to support this claim with experimental data from MNase-, DNase-, FAIRE-, or ATAC-seq, for example).
Author Response Manuscript changes	We fully agree that in order to make statements on chromatin accessibility, additional experiments would be required. Since this is a minor aspect of the manuscript and not required for the main message, we decided to remove this statement, and to focus on quantitative differences of H3K4me1 signal, being stronger at the FOXA1-bound enhancers relative to the ER-bound promoters. Line 194 add: In contrast to FOXA1-enriched sites, sites selectively occupied by ERα were weaker for active enhancer mark H3K4me1³⁸

Figure changes	(Supplementary Fig. 8A,B).’ Original Figure 2H-I has been moved to Supplemental Figure 8 and all references updated. Supplemental Figure 4 has been moved to Supplemental Figure 8 and all references are updated. Original supplemental figure 5 has been moved to supplemental figure 9 and references updated.
Reviewer 3 Minor issues comment 4	There is a lack of validation of their sequencing data by qPCR (for both ChIP-seq and RNA-seq datasets).
Author Response Manuscript changes	We agree with the reviewer that qPCR validations for ChIP-seq are necessary. We have provided these in a new supplement figure for ER, GR, AR and FOXA1 ChIPs, where we performed ChIP-QPCR analyses for the RARA enhancer, and correlated relative QPCR enrichment with normalized readcount from ChIP-seq for the same enhancer (Supplementary Figure S2). For RNA-seq validation, we would like to argue that the microarray-based expression data from Johanssen et al. (Figure 4A) is a suitable validation of our RNA-seq data. Line 144: added ‘(ChIP-seq validated by ChIP-QPCR in Supplementary Fig. 2)’ Line 369: added to Methods section: ‘ChIP-QPCR was performed to validate ChIP-seq data for ERα, GR, AR and FOXA1. For QPCR, relative enrichment of the RARA enhancer (chr:1738478661-38478809) (primers: GCTGGGTCCTGGCTGTTC (FWD) and CCGGGATAAAGCCACTCCAA (REV)) over a negative control region (primers: TGCCACACACCAGTGA CTTT (FWD) and ACAGCCAGAAGCTCCAAAA (REV)) was normalized over input, and plotted against the log(count/million) measurement from the same region from the MQ20 filtered aligned ChIP-seq file.’
Reviewer 3 Minor issues comment 5	Data for ChIP-seq and RNA-seq need to be deposited to a publically-available database such as GEO or ArrayExpress to allow others to use the data to inform their hypotheses
Author Response	We fully agree and have begun the process to deposit all sequencing data in GEO. We will provide a link to the data in the Methods section, and will make all ChIP-seq and gene expression publicly available upon acceptance of the paper.

Reviewer 3 Minor issues comment 6	Title should be re-worded. The manuscript does not really address the “Epigenetic Regulation” of SHRs, as only one histone mark (H3K4me1) was profiled and DNA methylation was not profiled. Maybe a better title would be “Characterizing steroid hormone receptor chromatin binding landscapes in male breast cancer”?
Author Response Manuscript changes	Thank you for the suggestion. With respect, we’d also like to add female to the title as we have now profiled 29 female samples for various hormone receptors, and substantially strengthen the female data throughout the manuscript. Title is changed to: “Characterizing steroid hormone receptor chromatin binding landscapes in male and female breast cancer”.
Reviewer 3 Minor issues comment 7	On lines 135-136, it is stated “ERa-regions are bound by all the factors”. This observed phenomenon is a first for MBC, but maybe the authors could cite that it was previously known different transcription factors ‘pile up’ with ERa in female breast cancer cells (e.g., with RAR, GATA3, FOXA1, etc.- Liu, Z. et al. 2014 Cell). Also, the Discussion section lines 260-263 seem to suggest ERa, AR, GR, and PR bind regions of DNA together due to closely juxtaposed EREs and ARE/GRE/PREs. However, a ‘tethering’ model whereby ERa might recruit some of the other SHRs (or vice versa) should be mentioned here as an alternative, given prior publications like Liu, Z. et al. that argue for such action.
Author Response Manuscript changes	Thank you for the suggestion. We have updated the text accordingly. Line 148 add: ‘but is reminiscent of FBC transcription factors such as GATA3, FOXA1, RARα, which have been found to be bound in the same regions⁴⁰.’ Line 290, add: ‘Alternatively, genomic overlap of SHR binding profiles may be the consequence of ‘tethering’, in which factors associate to the DNA indirectly through complex formation with DNA-bound factors⁴⁰, as was recently described for ERα and PR³⁰.’
Reviewer 3 Minor issues comment 8	How might knowing the 14 genes in Figure 5G be used in the clinic? For use as potential biomarkers? Could any of these be targeted by small molecule inhibitors (e.g., CAMKK2)?

Author Response	The positive or negative coefficient of a specific gene indicates expression of the gene is negatively or positively prognostic, respectively. Among them, indeed, CAMKK2 can be of interest, which is known as an AR target that regulate both cancer cells and macrophages (Racioppi et al. Trends Mol Med. 2013). As stated in the manuscript, the gene expression signature collectively stratifies patients by outcome. When tested independently, the 14 genes do not show the same predictive performance. With this, we envision the combined signature to have potential as a prognostic biomarker for male breast cancer outcome, analogous to mamaprint, Oncotype DX or PAM50 that are used in female breast cancers. Interestingly, a number of the 14 genes are targetable with existing small molecule inhibitors, such as CAMKK2 (STO-609, Hawley et al. Cell Met. 2005), CAPN9 (Chen et al. Exp. Cell Res. 2010) BACE2 (Canalis et al. Endocrinol. Metab. Clin. North Am. 1989) and TNFSF11 (aka RANKL) (Pageau, mAbs, 2009), and future studies may further elucidate whether one or more of these inhibitors would be applicable in the treatment of male breast cancer.
Manuscript changes	Line 311: add, ‘The union of genes under differential control of ER and FOXA1 jointly classify patients on outcome, and it remains to be determined which transcription factor is facilitated by FOXA1 at these sites. The 14 genes classifier we identified may be of added value as male breast cancer-specific prognostic classifier, but further validation of these results would be needed. Furthermore, small molecule inhibitors are available for a number of the 14 genes represented in the classifier, such as CAMKK2 (STO-609)⁵¹, CAPN9⁵², BACE2⁵³ and TNFSF11 (aka RANKL)⁵⁴, and future studies may further elucidate whether these inhibitors may be applicable in the treatment of male breast cancer.’
Reviewer 3 Minor issues comment 9, other edits for better clarity	Please note, the line numbers have changed in the revised manuscript. a) On lines 60-61 and 245, please change “enhancer-selective” to “enhancer-enriched” as H3K4me1 is often found at highly active promoter regions as well. b) On line 82 please add “(such as tamoxifen)” after “endocrine therapies”. c) On line 89, replace “form” with “from”; on line 227, please add “to” after “(Methods)”; and on line 254 replace “then” with “than”. d) Sentence on lines 93 and 94 starting with “AR activation in breast cancer cells...” cited references 19-21, but these studies used LNCaP

	prostate cancer cells, so please re-word this sentence or add different references. In fact, references 19-21 could be combined with references 22-25 later on line 95. e) Please add “in female breast cancer” after “ERa bound regions” on line 133. f) On line 139, it is not clear that “enhancer regions” were bound but that introns and distal intergenic regions were. Please change wording as one does not truly know these regions are all enhancing transcription. g) In Figure 3A, please use different colors for M1 and M2 than blue and red that are also used in the expression level scale. Also similar concern for Figure 3G (with green and purple representing both ChIP-seq versus RNA-seq as well as insulin signaling versus immune related) and Figure 5B (with red/blue for M1/M2 subtype as well as using these two colors for the heatmap color scale). h) On line 197-199, AR signaling was not the only pathway displaying better enrichment from ChIP-seq versus RNA-seq and thus this sentence needs re-wording. i) Why does Figure 5F show a p-value of 0.048, yet the figure legend (line 542) reports 0.041? j) Why are references 48 and 51 the exact same thing? Please delete one of these.
Author Response Manuscript changes	We thank this reviewer for so closely reading our manuscript. We have changed all suggestions. Please note, line numbers are now different from previously cited by the reviewer. All textual changes have been made. d) Sentence on lines 93 and 94 starting with “AR activation in breast cancer cells...” cited references 19-21, but these studies used LNCaP prostate cancer cells, so please re-word this sentence or add different references. In fact, references 19-21 could be combined with references 22-25 later on line 95. We have removed the references and added (Wang. Et al. Expression of androgen receptor and its association with estrogen receptor and androgen receptor downstream proteins in normal/benign breast luminal epithelium. Appl. Immunohistochem. Mol. Morphol. AIMM Off. Publ. Soc. Appl. Immunohistochem. 22, 498–504 (2014) and Peters et al. Androgen receptor inhibits estrogen receptor-alpha activity and is prognostic in breast cancer. Cancer Res. 69, 6131–6140 (2009).

g) In Figure 3A, please use different colors for M1 and M2 than blue and red that are also used in the expression level scale. Also similar concern for Figure 3G (with green and purple representing both ChIP-seq versus RNA-seq as well as insulin signaling versus immune related) and Figure 5B (with red/blue for M1/M2 subtype as well as using these two colors for the heatmap color scale).

We have changed the colors accordingly to avoid confusion.

i) Why does Figure 5F show a p-value of 0.048, yet the figure legend (line 542) reports 0.041?

Thank you for noticing this mistake. We have reported 0.048 in both the figure and the legend.

j) Why are references 48 and 51 the exact same thing? Please delete one of these.

We have removed the duplicate reference.

Reviewers' comments:

Reviewer #1 (Remarks to the Author):

Response to the Authors:

In the new version of the manuscript, the Authors have not addressed most of the points raised while evaluating the original version. Each of the individual molecular players analysed (ER, AR, PR, FOXA1 and GATA3) has been already investigated in female breast cancer patients. The Authors have expanded their study to male breast cancer tissues and compared the molecular differences between male and female. The study is comprehensive in relation of number of factors studied. However, the number of samples analysed is very low (≥ 3 samples/factor).

The study had a very ambitious initial aim. However, the conclusions might not be true due the low number of samples analysed for each of the factors investigated. The Authors conclude that their comprehensive overview reveals gender-selective and genomic location-specific hormone receptor action. Are the Authors considered the tumour heterogeneity in their analyses?

In summary, the revised manuscript provides still largely correlative evidence that may indicate some differences between male and female tissue. However, the conclusions of this study are not validated experimentally and derived from a very low number of male samples.

Major specific comments:

1) Figure 2A: the Authors have represented the overlap of ER α with the other steroid hormone receptors investigated or pioneer factors FOXA1 and GATA3. When Authors examine ER α , FOXA1 and GATA3 they identify unique binding sites for each of these factors (79% for FOXA1, 42% for ER and 33% for GATA3). This is a not a novel finding. Previously, it was reported in breast cancer cell lines the existence of unique sites for ER and FOXA1 (PMID: 18358809, PMID: 21151129, PMID: 23172872). Most of these studies come from Miles Brown and Jason Carroll laboratories. One of the conclusions from the all these studies was that not all the ER binding interactions overlap with FOXA1. However, the depletion of FOXA1 impacted the 95% of the ER chromatin interactions, which supports that the lack of physical overlap of two factors does not imply the lack of interplay between them. The genomic regulation of these factors is quite complex and usually chromatin is organized in a 3 dimensional way, which imply that factors that are physically distant with ER might be involved in the regulation of the same target genes (PMID: 19890323). Therefore, obtaining conclusions without experimental validation is highly risky. The Authors have not considered these factors before to make such conclusions. The Authors state that by means of using Pearson correlation of the consensus binding sites of all the factors investigated they observed FOXA1 profiles do not correlate with other profiles. The authors should perform the same analysis by comparing FOXA1 and ER binding sites rather than considering the overlap of all the other regions to claim that ER and FOXA1 do not correlate.

2) Figure 3: the Authors show that ER α sites devoid of FOXA1 were strongly promoter-enriched (Fig. 3A-C). Results that suggest to the Authors that the model systems currently used do not adequately capture the genomic distributions of ER α found in clinical samples and therefore it might suppose a novel finding in male breast cancer. This is an interesting finding, however is not novel. The same group have a publication (PMID: 22907427) where they described that in two different breast cancer cell lines the phosphorylation of ER α redirected the binding of the nuclear receptor to promoters.

3) Figure 4: the Authors find 1395 differential ER α DNA binding sites (Fig. 4B), which were all co-occupied with FOXA1, AR, GR, PR and GATA3 (Fig. 4C) and from these sites they search for genes with proximal binding sites (<20kb or within the gene body). Subsequently, they examine for molecular and biological associations using pathway analysis (IPA, Qiagen). Both ER α ChIP-seq associated M1/M2 genes and gene expression-based M1/M2 genes were strongly associated with ER α pathway indicators. Why the authors have not determined the binding of FOXA1 or GATA3

with M1/M2 genes?

4) Figure 5 and 6: The Authors investigate 30 male breast tumours (Fig. 5A), along with its pioneer factor FOXA1 (n=7 for both genders) (Fig. 5B). No clear differences in ER α and FOXA1 binding were found between genders, on the level of peak overlap ratio (Fig. 5A 5D) or relative read counts in peaks (Fig. 5E, 5F).

Given that the ER α sites in female tumours seemed not to be indicative for male patient outcome in the male BC series investigated in this study, the authors analysed the binding sites of ER α and pioneer the factor FOXA1 in MBC for outcome prediction. Based on lymph node status (Supplementary Figure 16), the authors can predict patient outcome. The authors also performed logistic regression to construct a supervised binary classification model to predict gene signatures that could predict outcome in male patients for both ER α - and FOXA1-based classifiers. By performing the mentioned analyses the authors identified a gene expression signature of 14 genes that synergistically classify male BC on outcome (Fig. 6F).

In summary, supplementary Fig. 16 is showing the overall survival of LN-negative and positive male patient and Fig. 6F the overall survival of the two groups of patients (high- and low-risk groups) driven from the classification model trained with union of potential targets. The Kaplan-Meier analysis based on LN status predicts a significant OS higher than 80% after 16 years of diagnosis (Fig. 6F) and the Kaplan-Meier analysis based on gene signature identified by the authors predicts a significant OS lower than 80% after 16 years of diagnosis (Supplementary Fig. 16). Based on the new results proposed by the Authors, why the new model might be a better tool compared to LN status to predict the survival of male breast cancer patients? It is accepted in clinic that the number of involved nodes is one of the most important prognostic factors for breast cancers in men to predict disease-specific survival (PMID: 8562148, PMID: 8416712).

5) The Authors propose the use in the future of small molecule inhibitors targeting some of the 14 genes represented in the male specific gene signature (i.e. CAMKK2 and others). In this regard, it has been shown in a registry study involving 829 men with primary breast in cancer (PMID: 23341341) that the 82%, 15%, and 4% had hormone receptor-positive, human epidermal growth factor receptor 2 (HER2)-positive, and triple-negative breast cancers, respectively. Importantly, these patients are treated with the same therapies as female are treated (surgery, neoadjuvant or adjuvant therapies). For adjuvant therapies targeting, it is recommended hormone therapy for patients with hormone receptor-positive breast cancer. This recommendation is based largely upon the benefits that have been observed in clinical trials performed in women and from retrospective studies that compared male and female treatment (PMID: 16270318, PMID 10091736). As in the adjuvant setting, tamoxifen is the preferred initial agent for men with metastatic breast cancer. Over 80% of men with hormone receptor-positive disease will respond to tamoxifen (PMID: 12379069), which represents a similar number described in female. For men who experience progression on or after tamoxifen, alternative endocrine therapies are available. Men that no respond to one form of hormonal treatment have a greater likelihood of responding to subsequent hormonal manipulations (PMID: 2933617), which does not differ substantially from the clinical results in female.

In conclusion, the revised manuscript is just a descriptive study that does not appear to provide direct and robust experimental evidence for making it a highly interesting and significant study in the area of breast cancer.

Reviewer #2 (Remarks to the Author):

The authors have been responsive to the reviewer's comments and the manuscript has improved as a result.

I only have a remaining comment:

As for the introduction, the change: "resistance is common" to "relapse after hormonal treatment

has been noted..." should be made also in the abstract.

Reviewer #3 (Remarks to the Author):

This revised manuscript by Severson, T. M. et al. now profiles the chromatin occupancy of four steroid hormone receptors (SHRs) (estrogen receptor alpha, ERa; androgen receptor, AR; glucocorticoid receptor, GR; and progesterone receptor, PR), two potential 'pioneer' factors (FOXA1 and GATA3, with the GATA3 ChIP-seq data being new as requested by Reviewer #3), and one enhancer-enriched histone modification (H3K4me1) in 49 male breast cancer (MBC) patient samples to begin to understand at a molecular level what may drive this rare breast cancer. The authors also now provide cistromic data for the same proteins from eight female breast cancer (FBC) patient samples isolated from their associated hospital and FBC-derived cell lines (MCF7, T47D, ZR75) and MCF7 xenograft tumors to allow a direct comparison with MBC. With these new data, the manuscript is more improved and a stronger candidate for publication in Nature Communications, although it remains largely descriptive. The authors have now provided key information about the patients and have largely addressed the major points raised during the previous review. Reviewer 1's concern that this study lacks any novelty has been addressed by observing that the majority of ERa chromatin binding sites in tumors of both sexes mainly occupy promoters not enhancers (Fig. 3c), in contrast to the dogma established from FBC cell lines. Reviewer 2's concern about small sample size has been adequately satisfied with 161 total ChIP-seq datasets now analyzed from a larger number of patients. Thus, the manuscript is almost acceptable as is and only needs some remaining issues to be addressed, as highlighted below.

Remaining issues:

1. The authors cite in their rebuttal and now in the Discussion section (lines 313-314) that the transcription factor that functionally collaborates with FOXA1 at its binding sites is unknown. Given the recent report that FOXA1 and the MLL3 histone methyltransferase functionally interact at enhancers in FBC cells (Jozwik KM et al. 2016 Cell Rep.), that MLL3 is a major 'writer' of H3K4me1 (based on seminal work from the laboratories of Kai Ge and Ali Shilatifard), and that H3K4me1 was more enriched at FOXA1-bound enhancers in this study, could it be that MLL3 plays a role with FOXA1 in MBC at these enhancers? Could some targeted MLL3 ChIP-qPCR be performed to address this?
2. The authors need to provide a published reference or new data in Fig. S3 that the GATA3 antibody used in their ChIP-seq data is 'specific' for its intended antigen.
3. Could the authors please address if GATA3 was predictive for survival in MBC? In the rebuttal, the authors suggested GATA3 might be taking over as a pioneer factor in MBC. It would be good for the authors to address in the text (in paragraph beginning on line 244) why they focused more on FOXA1 versus GATA3 for the survival analysis.

4. Edits for better clarity:

a) Textual changes:

In the Introduction (page 4), please change "ERa/DNA binding" to "ERa-DNA binding". Same change requested for "AR/DNA binding" on line 98. Please change "most extensively studies" to "most extensively studied" on line 97. On line 122, please add "cistrome" before "profiling". On line 173, please add "and GR" after "AR". On line 190, please replace "There" with "These". On line 227, please change "(Fig. 5A 5D)" to "(Fig. 5C, 5D)". On line 264, please remove the word "synergistically" as no test was indicated for synergy (e.g., Bliss independence). Please reduce the use of "epigenetic" and "epigenetics" in the text (e.g., line 115). As cited in last review, only one histone mark was profiled and no DNA methylation was assayed. These are truly "epigenetic" in

nature- not a transcription factor cistrome. References 62 and 65 are duplicated, so please remove one of them. Please remove "NCT02353988" from Discussion section on line 296 as this clinical trial was for AR-expressing, ER negative breast cancer that is not relevant to this present study.

b) Figures:

In Fig. 2b, it is hard to tell "light green" versus "dark green" to specify "FOXA1" versus "GATA3" as the figure legend states. Can the authors please use a more differential color scheme? In Fig. 4c, why is PR ChIP not shown in M1 samples, as text cited it on lines 208-209? Is PR M2 site specific? Also, M2 samples lacked FOXA1 and GATA3 ChIP- was this real or was the data just left off by mistake? Also, in Fig. 6e could the p-values be presented in a larger font?

c) Supplementary tables:

Please define in Sup Table 2 what the numbers below "ER" and "PR" indicate- maybe IHC staining intensity? Also, in Sup Table 3, please define what "PG" is.

RESPONSE LETTER

-- Reviewer 1 --

We would like to thank the referee for critically reading our rebuttal and updated manuscript as well as for the insightful comments. We were disappointed and surprised to read the reviewer finds the study too small although it is the biggest ChIP-seq report in breast tumors to date, not validated (while multiple validations are provided) and finds that we did not address most of the raised comments in our previous 25-page rebuttal, in which we highlighted major differences in male breast cancer and female breast cancer in relation to cell line studies. We sincerely hope that the reviewer finds the arguments laid out below reasonable and compelling in favor of acceptance of our work.

Reviewer 1 Comment 1	In the new version of the manuscript, the Authors have not addressed most of the points raised while evaluating the original version. Each of the individual molecular players analysed (ER, AR, PR, FOXA1 and GATA3) has been already investigated in female breast cancer patients. The Authors have expanded their study to male breast cancer tissues and compared the molecular differences between male and female. The study is comprehensive in relation of number of factors studied. However, the number of samples analysed is very low (≥ 3 samples/factor). The study had a very ambitious initial aim. However, the conclusions might not be true due the low number of samples analysed for each of the factors investigated. The Authors conclude that their comprehensive overview reveals gender-selective and genomic location-specific hormone receptor action. Are the Authors considered the tumour heterogeneity in their analyses? In summary, the revised manuscript provides still largely correlative evidence that may indicate some differences between male and female tissue. However, the conclusions of this study are not validated experimentally and derived from a very low number of male samples.
Author Response	We respectfully disagree with the reviewer that we did not address most of the points originally raised. We feel that our detailed 25-page point-by-point rebuttal addressed each question of all reviewers extensively. We would like to stress again that only ERα has been chipped before on clinical tissue (PMID 22217937), as highlighted in our previous rebuttal

Manuscript changes	as well. To date, no report has described ChIP-seq for AR, PR, GR, FOXA1, GATA3 or H3K4me1 in any human breast tumors, male or female. Our pioneering assessment of ChIP-seq for AR, PR, GR, FOXA1, GATA3 and H3K4me1 in human breast tumors not only confirms parts of prior cell line-based studies, but also reveals features that are unique for tumors that were not found in cell lines, namely the promoter-enrichment of FOXA1-devoid ER sites . On several occasions, the reviewer mentions the low numbers of samples in our study. We have now added a total of 161 ChIP-seq datasets for both male and female breast cancer, 110 of which are from clinical specimens. This is an unprecedented number of tumor specimen ChIP-seq samples in a single study and greatly exceeds the number of tumors reported by other labs that have produced a large quantity of ChIP-seq data such as the Carrol lab in breast cancer (PMID 22217937: n=18 breast tumors, only ER ChIP, Nature) and the Neal lab (PMID 23260764 : n=12 prostate tumors, only AR ChIP, Cancer Cell). The reviewer cites from the manuscript (original line 131) that we merely show ≥ 3 samples/factor. This number in the text is indeed confusingly low, since we successfully performed 30 ERα, 7 FOXA1, 10 AR, 7 GR and 4 PR ChIP-seq experiments on male breast cancer samples. For females, we show 29 ERα and 8 FOXA1 samples. GATA3 ChIP-seq was added in response to reviewer#3, which is the sole factor for which we only able to obtain 3 ChIP-seq datasets. We agree this is not clear in the manuscript and we have adjusted the sentence in the manuscript (lines 131 - 134) to inform the reader. Furthermore, the data are consistent between previous analyses (lower sample numbers) and with the addition of new ChIP-seq samples, indicating our findings are robust. Note that the number of tumor samples analysed in this study by ChIP-seq is far larger than the number of ER-positive breast cancer cell lines available, while high-ranking journals request the use of at least 2 cell lines to confirm findings. We show results for ~60 tumors. We have considered the possibility that tumor heterogeneity may be a confounding factor in the gender-selectivity of ERα and FOXA1. Figure 5C/D shows that the peak overlap ratio between male and female samples is similar, indicating that the level of heterogeneity is equivalent between gender and given the robustness of the signal, sample numbers are sufficient to support our conclusions. To address
---

this point further, we now provide data from one tumor, of which two separate sections were processed independently. As shown in the new supplementary figure 2, concordance of signal between both experiments was very high (Pearson correlation=0.82). See below.

In addition, we performed an alternative analysis to investigate the impact of potential differences in tumor heterogeneity between genders. Previously, variant allele frequency has been shown to be associated with tumor percentage in bulk tumor sequencing analyses (PMID 26919633, Figure 4). To investigate in our data, we called variants in the ER ChIP-seq data for both male and female tumor sample cohorts in the manuscript (samtools, chromosome 10). We filtered for read coverage across the variants of at least 50 and only heterozygous variants. We then took each sample's average variant allele frequency and compared the distribution between the two groups. In a perfect situation, where the complete tumor sample has a tumor percentage of 100%, the variant allele and reference allele of a

heterozygous variant percentage should be 50% and 50%, respectively. We identified no significant difference in distribution between the two groups ($p=0.54$, Wilcoxon rank sum) (see below). Combined, these analyses rule out heterogeneity as a major factor of bias in our conclusions about differences between male and female cohorts.

Furthermore, to focus our analyses on the most-robust peaks and thus minimizing potential impact of heterogeneity, we only considered peaks that were found in around 50% or more of the tumors, which is very conservative compared to (some) work which this reviewer cites (PMID: 22217937), in which all peaks found in at least 2 out of 21 tumors were considered. We have incorporated a statement on this in the results section (lines 308 - 314).

We disagree with the statement of the reviewer that the conclusions from our study are not validated. Our conclusions have been extensively validated with ChIP-qPCR (Supplementary Figure 3) and antibody specificity experiments (Supplementary Figure 4). Furthermore, the additional analysis of correlation between read count in peaks in two independent ERa chips performed on two separate sections of tumor from a male patient (new Supplemental Figure 2), represents independent biological validations. Also in the gene

	expression analyses and subsequent patient clustering (Figure 6B/C), we provide data from a discovery and validation cohort. Cumulatively, we strongly feel that the data are thoroughly validated on multiple levels. The study is mainly a genomics-driven clinical analysis and consequently correlative per definition; just like any other clinical study. Since male breast cancer model systems simply do not exist and as our data now shows, female model systems do not comply with actual clinical tissue, we strongly believe that also correlative studies such as ours are highly novel, relevant and significantly contribute to the field.
Reviewer 1 Comment 2	Major specific comments: 1) Figure 2A: the Authors have represented the overlap of ERα with the other steroid hormone receptors investigated or pioneer factors FOXA1 and GATA3. When Authors examine ERα, FOXA1 and GATA3 they identify unique binding sites for each of these factors (79% for FOXA1, 42% for ER and 33% for GATA3). This is a not a novel finding. Previously, it was reported in breast cancer cell lines the existence of unique sites for ER and FOXA1 (PMID: 18358809, PMID: 21151129, PMID: 23172872). Most of these studies come from Miles Brown and Jason Carroll laboratories. One of the conclusions from the all these studies was that not all the ER binding interactions overlap with FOXA1. However, the depletion of FOXA1 impacted the 95% of the ER chromatin interactions, which supports that the lack of physical overlap of two factors does not imply the lack of interplay between them. The genomic regulation of these factors is quite complex and usually chromatin is organized in a 3 dimensional way, which imply that factors that are physically distant with ER might be involved in the regulation of the same target genes (PMID: 19890323). Therefore, obtaining conclusions without experimental validation is highly risky. The Authors have not considered these factors before to make such conclusions. The Authors state that by means of using Pearson correlation of the consensus binding sites of all the factors investigated they observed FOXA1 profiles do not correlate with other profiles. The authors should perform the same analysis by comparing FOXA1 and ER binding sites rather than considering the overlap of all the other regions to claim that ER and FOXA1 do not correlate.

Author Response	The reviewer states that the overlap of steroid hormone receptors with pioneering factors is not a novel finding based on cell line data. We acknowledge that a level of overlap has been previously reported in cell lines, but we now show that in tumor samples the genomic distribution of ER-enriched sites is completely different from cell lines and Forkhead motifs are absent (Figure 3 and supplementary figure 10). These findings are completely novel and especially relevant for therapeutics development relying on these cell line models to reflect the patient situation. The reviewer states that obtaining conclusions without experimental validation is highly risky. We would argue that it is equally risky to base conclusions and dogmas exclusively on cell lines without further validation in clinical specimens, especially since we now show that the genomic behavior of ER in relation to FOXA1 is not conserved between cell lines and the real human tumor condition (Figure 3). Again, we note that our findings have been extensively experimentally validated on the level of antibody specificity, ChIP-qPCR, are consistent between the tumors we study, and reveal consistent patient clustering based on expression data in two independent male breast cancer cohorts. We believe all the above-mentioned arguments indicate the robustness of our findings. Please note this is intrinsically a genomics-driven clinical study, where we show steroid hormone receptor chromatin binding profiles in a setting for which no model system exists (male breast cancer) and we now show that model systems are not compatible with the actual human disease (female breast cancer). With that, experimental validation of our findings in a suitable model system is currently not possible, since none exists. The reviewer asks us to perform a pearson correlation matrix with binding sites of only FOXA1/ER. This comparison can be found in figure 2E, already presented in the previous submission. This figure demonstrates FOXA1 and ER binding sites cluster separately in the pair-wise correlation matrix.
Reviewer 1 Comment 3	Major specific comments: 2) Figure 3: the Authors show that ERα sites devoid of FOXA1 were strongly promoter-enriched (Fig. 3A-C). Results that suggest to the Authors that the model systems currently used do not adequately capture the genomic distributions of ERα found in clinical samples and therefore it might suppose a novel finding in male breast cancer. This is

	an interesting finding, however is not novel. The same group have a publication (PMID: 22907427) where they described that in two different breast cancer cell lines the phosphorylation of ERα redirected the binding of the nuclear receptor to promoters.
Author Response	The reviewer may have misunderstood the differences between both reports, which we are happy to clarify. The current study interrogates total ER in tumors, which we compare with total ER in cell lines. Our results show a strong deviation in the observations between both settings which suggests that current cell line models may not be appropriately for modelling some ER processes in tumors. The manuscript to which the reviewer refers describes a very specific phosphorylation of ER at serine residue 305, which represents only a very small subset of ER molecules (Michalides et al., 2002 Cancer Cell. PMID: 15193262). In treatment-naïve primary breast cancers, only a low percentage of tumor cells are positive for ERαS305P, and around 20% of primary breast cancers is positive for this phosphorylation (Holm et al., 2009 J Pathology. PMID:18991335). These findings are not relevant in the current setting as we have performed CHIP-seq for bulk tumor/ER, which is the design of typical CHIP-seq analyses.
Reviewer 1 Comment 4	3) Figure 4: the Authors find 1395 differential ERα DNA binding sites (Fig. 4B), which were all co-occupied with FOXA1, AR, GR, PR and GATA3 (Fig. 4C) and from these sites they search for genes with proximal binding sites (<20kb or within the gene body). Subsequently, they examine for molecular and biological associations using pathway analysis (IPA, Qiagen). Both ERα CHIP-seq associated M1/M2 genes and gene expression-based M1/M2 genes were strongly associated with ERα pathway indicators. Why the authors have not determined the binding of FOXA1 or GATA3 with M1/M2 genes?
Author Response	Between the M1 and M2 male breast cancer subtypes, which are based on gene expression, the patients are not equally distributed. With this, only the ER CHIP-seq data was sufficiently powered to have enough patients to sufficiently represent both M1 and M2 subgroups, as was apparent in Figure 4 where we show distribution of all other factors over M1 and M2 patient subpopulations.
Manuscript changes	This is now better explained in the manuscript (Lines 211 - 213).

Reviewer 1 Comment 5	4) Figure 5 and 6: The Authors investigate 30 male breast tumours (Fig. 5A), along with its pioneer factor FOXA1 (n=7 for both genders) (Fig. 5B). No clear differences in ERα and FOXA1 binding were found between genders, on the level of peak overlap ratio (Fig. 5A 5D) or relative read counts in peaks (Fig. 5E, 5F). Given that the ERα sites in female tumours seemed not to be indicative for male patient outcome in the male BC series investigated in this study, the authors analysed the binding sites of ERα and pioneer the factor FOXA1 in MBC for outcome prediction. Based on lymph node status (Supplementary Figure 16), the authors can predict patient outcome. The authors also performed logistic regression to construct a supervised binary classification model to predict gene signatures that could predict outcome in male patients for both ERα- and FOXA1-based classifiers. By performing the mentioned analyses the authors identified a gene expression signature of 14 genes that synergistically classify male BC on outcome (Fig. 6F). In summary, supplementary Fig. 16 is showing the overall survival of LN-negative and positive male patient and Fig. 6F the overall survival of the two groups of patients (high- and low-risk groups) driven from the classification model trained with union of potential targets. The Kaplan-Meier analysis based on LN status predicts a significant OS higher than 80% after 16 years of diagnosis (Fig. 6F) and the Kaplan-Meier analysis based on gene signature identified by the authors predicts a significant OS lower than 80% after 16 years of diagnosis (Supplementary Fig. 16). Based on the new results proposed by the Authors, why the new model might be a better tool compared to LN status to predict the survival of male breast cancer patients? It is accepted in clinic that the number of involved nodes is one of the most important prognostic factors for breast cancers in men to predict disease-specific survival (PMID: 8562148, PMID: 8416712).
Author Response	It should be noted that the cohort in Figure 6F is our validation cohort for the predictive gene signature. This cohort was divided into low and high-risk groups based on the gene signature and significantly classified patients on distant-metastasis-free survival (not overall survival as mentioned by the reviewer). The Kaplan Meier in Figure 6F is not based on LN status, but is based on a gene expression signature designed to capture predictive features. This signature retains significance (threshold $p < 0.01$) even when LN status is included in the multivariate model. This indicates it has a predictive capacity above and beyond LN status. The discovery cohort that was used to predict patient outcome (overall survival) based on LN status, is represented in supplementary Figure

	17. With our gene expression signature we aimed to produce a biomarker for outcome prediction. Such a signature provides more information than the binary measure LN status. For example, the genes found in this classifier might be putative targets for therapy, which would not have been identified otherwise. In addition, we found our classifier performs best using both ER and FOXA1 to classify patients on outcome, suggesting there may even be additional features worth targeting. Most relevantly, with this study we aim to improve our understanding of the biology of male and female breast cancer, using existing and known clinical parameters as starting point. This approach is not per se aimed to replace existing clinical parameters, but meant to complement these and to learn more about the disease, analogous to the pioneering paper of Perou and Sorlie (Nature, 2000) or the TCGA paper on breast cancer (TCGA, 2012, Nature).
Reviewer 1 Comment 6	5) The Authors propose the use in the future of small molecule inhibitors targeting some of the 14 genes represented in the male specific gene signature (i.e. CAMKK2 and others). In this regard, it has been shown in a registry study involving 829 men with primary breast in cancer (PMID: 23341341) that the 82%, 15%, and 4% had hormone receptor-positive, human epidermal growth factor receptor 2 (HER2)-positive, and triple-negative breast cancers, respectively. Importantly, these patients are treated with the same therapies as female are treated (surgery, neoadjuvant or adjuvant therapies). For adjuvant therapies targeting, it is recommended hormone therapy for patients with hormone receptor-positive breast cancer. This recommendation is based largely upon the benefits that have been observed in clinical trials performed in women and from retrospective studies that compared male and female treatment (PMID: 16270318, PMID 10091736). As in the adjuvant setting, tamoxifen is the preferred initial agent for men with metastatic breast cancer. Over 80% of men with hormone receptor-positive disease will respond to tamoxifen (PMID: 12379069), which represents a similar number described in female. For men who experience progression on or after tamoxifen, alternative endocrine therapies are available. Men that no respond to one form of hormonal treatment have a greater likelihood of responding to subsequent hormonal manipulations (PMID: 2933617), which does not differ substantially from the clinical results in female.

Author Response	In the last review round, reviewer 3 requested us to further elaborate on the 14 genes we identified, and whether any of these would represent potential drug targets in this setting. As response to this reviewer's request we now further elaborated on these 14 genes. For a number of these factors, including CAMKK2, small molecule inhibitors have been generated and these may be of interest for further studies in this setting. As we extensively describe in the introduction, current treatment modalities of male breast cancer are indeed based on the female disease. Even though there are many similarities between male and female breast cancer as rightfully pointed out by the reviewer, nonetheless, male patients may benefit (even) more from therapies that target processes intrinsic to male rather than female breast cancer. With the current setup of the introduction section, we believe we sufficiently appreciate the literature as mentioned by the reviewer at this point.
Reviewer 1 Comment 7	In conclusion, the revised manuscript is just a descriptive study that does not appear to provide direct and robust experimental evidence for making it a highly interesting and significant study in the area of breast cancer.
Author Response	We strongly disagree with the reviewer in this aspect, in which the novelty of our work was also acknowledged by the other two reviewers. This manuscript is a pioneering study, for the first time describing genome-wide chromatin binding profiles of 4 steroid hormone receptors, 2 pioneer factors and a histone modification in clinical tissue from the exceptionally rare disease male breast cancer, as well as female breast tumors. In addition, we feel the study provides robust and experimental evidence that features believed key to ER genomics (being enhancer-enriched in cell lines) are not recapitulated in actual human tumors. Being intrinsically a genomically-driven clinical analysis, and thus per definition descriptive, mechanistic analyses are not feasible. This is especially relevant since we show here that for some features of ER cistromics, existing model systems (needed for perturbation of the system; a vital step for studying mechanisms) are not representing human tumors; we believe this is a highly novel and significant finding and we hope the reviewer would agree with us.

-- Reviewer 2 --

We would like to thank the referee for critically reading of our rebuttal and updated manuscript. We are delighted to hear the reviewer finds the manuscript improved and no additional analyses or experiments are requested.

Reviewer 2 Comment 1	The authors have been responsive to the reviewer's comments and the manuscript has improved as a result. I only have a remaining comment: As for the introduction, the change: "resistance is common" to "relapse after hormonal treatment has been noted..." should be made also in the abstract.
Author Response Manuscript changes	We are very pleased we were able to address this reviewers concerns with our efforts and appreciate the opportunity to improve our manuscript. We have changed the sentence in the abstract to reflect the change in the introduction. Line 53 change: 'resistance is common' to 'relapse after hormonal treatment is also noted'.

-- Reviewer 3 --

We would like to thank the referee for critically reading our rebuttal and updated manuscript and for the constructive suggestions. We are delighted to hear the reviewer finds the manuscript almost acceptable as is and we sincerely hope the reviewer finds our answers to the newly raised points satisfactory.

Reviewer 3 Comment 1	Thus, the manuscript is almost acceptable as is and only needs some remaining issues to be addressed, as highlighted below. Remaining issues: 1. The authors cite in their rebuttal and now in the Discussion section (lines 313-314) that the transcription factor that functionally collaborates with FOXA1 at its binding sites is unknown. Given the recent report that FOXA1 and the MLL3 histone methyltransferase functionally interact at enhancers in FBC cells (Jozwik KM et al. 2016 Cell Rep.), that MLL3 is a major ‘writer’ of H3K4me1 (based on seminal work from the laboratories of Kai Ge and Ali Shilatifard), and that H3K4me1 was more enriched at FOXA1-bound enhancers in this study, could it be that MLL3 plays a role with FOXA1 in MBC at these enhancers? Could some targeted MLL3 ChIP-qPCR be performed to address this?
Author Response	We are very pleased our efforts in response to the reviewers’ comments were fruitful and that we have had an opportunity to improve our manuscript. Thank you for bringing MLL3 to our attention. We fully agree that it would be highly interesting to see whether MLL3 plays a role on FOXA1 in male breast cancer. However, since male breast cancer is a very rare disease and tissue available for research is quite limited, we have exhausted all available tissue in the previous review round when the reviewer requested GATA3 analyses, which we successfully provided. Therefore, no tissue is left for additional experimentation on MLL3 ChIP-qPCR. What we did however test whether MLL3/FOXA1 overlap is expected in this setting. Taking MLL3 ChIP-seq data from MCF-7, we examined MLL3 binding signal in FOXA1 sites for MCF-7, female and male (see below). In general there are high MLL3 read counts in FOXA1 sites regardless of system, indicating MLL3 plays a role with FOXA1 not just in MCF-7 but also in male and female breast tumors. However, it

	would require further validation on a genome-wide basis which we feel is beyond the scope of this manuscript.   MCF 7 FOXA1 sites    Female FOXA1 sites    Male FOXA1 sites   
Reviewer 3 Comment 2	2. The authors need to provide a published reference or new data in Fig. S3 that the GATA3 antibody used in their ChIP-seq data is ‘specific’ for its intended antigen.
Author Response Manuscript change	We apologize for this oversight. This GATA3 antibody has been used by the ENCODE consortium in their studies and the antibody was validated according to standard ENCODE guidelines. This information and a reference to the ENCODE guidelines and validation data is now included in the manuscript. We also thank this reviewer for bringing this to our attention as we also noticed that there was no information in the methods section about the antibody used. We have also added this information. We have added the references for the GATA3 antibody and validation documents to the reference list.
Reviewer 3 Comment 3	3. Could the authors please address if GATA3 was predictive for survival in MBC? In the rebuttal, the authors suggested GATA3 might be taking over as a pioneer factor in MBC. It would be good for the authors to address in the text (in paragraph beginning on line 244) why they focused more on FOXA1 versus GATA3 for the survival analysis.
Author	We would like to thank the reviewer for this perspective. We focused

Response Manuscript changes	on FOXA1 for the survival analysis for two reasons 1) GATA3 was not originally in the manuscript and was added as a response to a reviewer suggestion and 2) the number of GATA3 samples for ChIP-seq was very small (n=3) and we used the differentially bound regions to construct the classifier. We have added a sentence on line 254 - 255 to clarify this to the reader. In order to investigate the question posed by the reviewer, we have investigated GATA3 gene expression levels in relation to lymph node status. We observed no correlation, indicating that in this dataset GATA3 expression is not predictive of survival (Wilcoxon rank sum, $p=0.4686$). The figure below shows normalized gene expression values versus lymph node metastasis status, true or false). Reviewer 3 Comment 4	4. Edits for better clarity: a) Textual changes: In the Introduction (page 4), please change “ERa/DNA binding” to “ERa-DNA binding”. Same change requested for “AR/DNA binding” on line 98. Please change “most extensively studies” to “most extensively studied” on line 97. On line 122, please add “cistrome” before

	“profiling”. On line 173, please add “and GR” after “AR”. On line 190, please replace “There” with “These”. On line 227, please change “(Fig. 5A 5D)” to “(Fig. 5C, 5D)”. On line 264, please remove the word “synergistically” as no test was indicated for synergy (e.g., Bliss independence). Please reduce the use of “epigenetic” and “epigenetics” in the text (e.g., line 115). As cited in last review, only one histone mark was profiled and no DNA methylation was assayed. These are truly “epigenetic” in nature- not a transcription factor cistrome. References 62 and 65 are duplicated, so please remove one of them. Please remove “NCT02353988” from Discussion section on line 296 as this clinical trial was for AR-expressing, ER negative breast cancer that is not relevant to this present study.
Author Response	We thank the reviewer for the suggested textual edits, which all significantly improve the clarity of the paper. We appreciate the thorough and detailed feedback.
Manuscript changes	On line 115 (previously line 122) we have added “(using off targeting sequencing reads)” to clarify our methodology on copy number profiling. All other text edits have been incorporated as suggested.
Reviewer 3 Comment 3	b) Figures: In Fig. 2b, it is hard to tell “light green” versus “dark green” to specify “FOXA1” versus “GATA3” as the figure legend states. Can the authors please use a more differential color scheme? In Fig. 4c, why is PR ChIP not shown in M1 samples, as text cited it on lines 208-209? Is PR M2 site specific? Also, M2 samples lacked FOXA1 and GATA3 ChIP- was this real or was the data just left off by mistake? Also, in Fig. 6e could the p-values be presented in a larger font?
Author Response	As suggested by the reviewer, the colorcode has now been changed for the GATA3 figure legends in Figure 2b, making the differences in colorscheme for FOXA1 and GATA3 more prominent. While updating this figure, we noticed a mistake in the number of PR sites mentioned in figure 2c. This mistake has been corrected and we apologise for any confusion. For the analyses shown in Figure 4C, the M1/M2 allocation of the samples was not known at the onset of the experiments.

	Consequently, for some ChIPped factors only M2 samples (such as PR) or M1 samples (such as FOXA1 and GATA3) were analysed. It is important to stress that no data were left out nor excluded in any way. Do note, that even though M1 and M2 samples are not available for all ChIP datastreams, selectivity was clearly observed for the genomic regions separating M1 and M2, based on differential ER binding. This highlights the conclusion that all studied factors followed the differential binding between M1 and M2 as observed for ER.
Reviewer 3 Comment 3	c) Supplementary tables: Please define in Sup Table 2 what the numbers below “ER” and “PR” indicate- maybe IHC staining intensity? Also, in Sup Table 3, please define what “PG” is.
Author Response	We apologise for being unclear in this regard. The numbers below ER and PR in Sup Table 2 indicate percentage of tumor cells which stain positively. PG in Sup Table 3 stands for ‘Progesterone ‘. Both issues are now better explained/labeled in the table.
Manuscript changes	Percentage included in Sup Table 2, PG explained in Sup Tab 3.

Reviewer #2 (Remarks to the Author):

This revision has addressed my previous concerns and suggestions.

Reviewer #3 (Remarks to the Author):

This re-revised manuscript by Severson, T. M. et al. addresses the vast majority of this reviewer's issues. Thus, the manuscript is acceptable after only three very minor revisions are made (listed below).

Author reply: We are delighted to hear the reviewer find the manuscript acceptable and would like to thank the reviewer for the constructive comments. The revisions have all been incorporated as addressed below.

Remaining issues:

1. This reviewer completely understands the lack of material to perform targeted MLL3 ChIP-qPCR as asked for last time. However, after the sentence ending with "facilitated by FOXA1 at these sites" (lines 328-329), could the authors please add one sentence that the MLL3 histone methyltransferase may represent one candidate based on the published FOXA1 and MLL3 interaction in FBC cells (Jozwik KM et al. 2016 Cell Rep.), which will be tested in future studies?

Author reply: as suggested by the reviewer, we now included a sentence stating 'The MLL3 histone methyltransferase may represent one candidate to be tested in future studies based on the published FOXA1 and MLL3 interaction in FBC cells⁴⁷.' (line 376)

2. Can the authors please add "due to small (n=3) sample size" after "insufficient power" on line 256?

author reply: 'due to small (n=3) sample size' is now included

3. Edits for better clarity: On line 123, please add "cistrome" before "profiling". In Fig. 6e could the p-values be presented in a larger font?

Author reply: Both edits have been incorporated, as suggested